# Synthesis of Novel Phosphorus-Containing Derivatives of 1,3,4-Trimethylglycoluril via the Birum–Oleksyszyn Reaction

**DOI:** 10.3390/ijms242317082

**Published:** 2023-12-03

**Authors:** Sergey I. Gorbin, Abdigali A. Bakibaev, Vera P. Tuguldurova, Andrey V. Kotov, Gleb O. Sysoev, Andrei S. Potapov, Dmitry I. Pavlov, Victor S. Malkov, Alexey S. Knyazev, Dmitriy A. Kurgachev, Mark V. Michalchenkov

**Affiliations:** 1Department of Chemistry, Tomsk State University, 36, Lenin Avenue, Tomsk 634050, Russia; 2Nikolaev Institute of Inorganic Chemistry, 3 Lavrentiev Avenue, Novosibirsk 630090, Russia

**Keywords:** Birum–Oleksyszyn reaction, 1,3,4-trimethylglycoluril, aryldiphenylphosphonates, diastereomers

## Abstract

This work presents the synthesis of a new compound, 1-[aryl-(diphenylphosphono)methyl]-3,4,6-trimethylglycolurils, via the interaction of benzaldehyde and its mononitro- and monohydroxyderivatives with 1,3,4-trimethylglycoluril and triphenylphosphite. By varying the reaction conditions and the catalysts, the obtained product yields ranged from satisfactory to good. The diastereomers formed during the reaction were separated by semipreparative HPLC on the C18 stationary phase. The isolated diastereomers were characterized by ^1^H, ^13^C, and ^31^P NMR, and the structures of the diastereomers were confirmed using a single-crystal X-ray crystal structure analysis and quantum chemical calculations.

## 1. Introduction

Glycoluril derivatives have attracted the attention of researchers due to their wide range of applications and, above all, due to their biological activities, particularly their nootropic, neurotropic, anxiolytic [1,2], and antibacterial properties [3,4]. Additionally, glycoluril derivatives are widely used as “building blocks” to synthesize supramolecular compounds such as bambus[n]urils [5,6], cucurbit[n]urils [7,8], and molecular clips [9,10]. Special attention has been paid to studying methods for synthesizing new glycoluril derivatives and their structural analogs and precursors as well as to the characterization of their properties [11,12,13,14,15]. Individual phosphorus-containing glycoluril derivatives are described in the literature [16,17], where their catalytic activity in some three-component reactions has been shown. Phosphorylated glycoluril derivatives were investigated as flame-resistant additives for rubber [18] and as moleculars tweezer exhibiting high binding properties to aliphatic amines [19]. However, these reports are sporadic, and to date, only a relatively small number of synthetic approaches have been developed for preparing phosphorus-containing glycoluril derivatives [16,17,18,19], and the main regularities of the formation of stereoisomers of the phosphonate derivatives of glycoluril have not been established either. We previously found [20] that the interaction of 1,3,4-trimethylglycoluril with aliphatic aldehydes and triphenylphosphite in acetonitrile using methanesulfonic acid as a catalyst led to the formation of 1-[1-(diphenylphosphono)alkyl]-3,4,6-trimethylglycoluril.

## 2. Results and Discussion

### 2.1. The Effect of Synthesis Conditions on the Yield of Phosphonate Derivatives of 1,3,4-Trimethylglycoluriles

In order to extend the methods for synthesizing phosphorus-containing glycoluril derivatives, we studied the conditions for the synthesis of various new glycoluril phosphonates **4a**–**f** using the Birum–Oleksyszyn approach exemplified by 1,3,4-trimethylglycoluril and benzaldehydes **2a**–**2f** (Figure 1).

It is noteworthy that the attempt to synthesize phosphonate derivatives of 1,3,4-trimethylglycoluryl **4a** was unsuccessful under typical conditions used to prepare diphenyl 1-aminophosphonates [21], namely, with glacial acetic acid as a solvent. Thus, according to the NMR analysis of the reaction mixture, the triphenylphosphite **3** interaction with benzaldehyde **2a** in glacial acetic acid with 1,3,4-trimethylglycoluril **1** (80 °C, 2 h) led to the formation of target compound **4a** with a low yield of 21%. In addition, the ^31^P NMR spectra of the reaction mixture contained clear signals within 3.40–−0.57 ppm, indicating both partial and complete hydrolysis of the ester groups for compound **3** as well as the formation of α-hydroxyphosphonate **6a** (^31^P δ = 15.99 ppm) as a by-product (Figure 2) (Appendix A).

The authors carried out the selection of catalysts, solvents, temperatures, and reaction times to establish the optimal conditions for synthesizing the target product **4a** because the typical conditions for the synthesis of α-aminophosphonates proved to be inefficient in obtaining phosphonate derivatives of 1,3,4-trimethylglycoluryl **1**. 

A series of model syntheses of compound **4a** was carried out to select the optimal reaction conditions considering the fact that Lewis acids and organic Brønsted acids tend to show a catalytic effect in the Birum–Oleksyszyn reaction [22,23]. Therefore, the reaction was carried out in anhydrous acetonitrile; AlCl_3_, BF_3_·Et_2_O, SnCl_4_, and TiCl_4_ were used as Lewis acids, whereas trifluoroacetic acid, methanesulfonic acid, and trifluoromethanesulfonic acid were used as Brønsted acids in addition to glacial acetic acid (Table 1). The amount of catalyst equaled 10 mol% of the total mass of the reaction mixture, and the process was carried out at room temperature. The reaction flow was monitored by HPLC.

Table 1 shows that the use of glacial acetic acid as a solvent resulted in a relatively low yield of the target product **4a** (23%). The maximum concentration of the latter in the reaction mixture was reached within 60 min according to HPLC (Table 1, entry 10). However, with glacial acetic acid as a catalyst and acetonitrile as a solvent, the content of **4a** remained relatively low, although it increased with time (Table 1, entries 1 and 2).

If trifluoroacetic, methanesulfonic, and trifluoromethanesulfonic acids (i.e., organic acids stronger than acetic acid) are applied as catalysts, the content of the target product **4a** in the reaction mixture increases by up to nine times (Table 1, entries 3, 4, and 5). When trifluoroacetic acid is applied within a comparable reaction time, the analytical yield of **4a** becomes almost six times higher than in the case of synthesis with glacial acetic acid. 

For methanesulfonic acid, the maximum yield for trimethylphosphonate glycoluril **4a** equaled 91% (82% isolated preparatively). This yield was achieved within 30 min under the selected reaction conditions. An increase in the reaction time when using methanesulfonic acid led to a decrease in the concentration of the target product **4a**, which might indicate hydrolysis and the formation of the reaction by-products.

The application of trifluoromethanesulfonic acid also leads to a notable increase in the content of the target substance **4a** for a relatively short reaction time (Table 1, entry 5), but the analytical yield of compound **4a** is only 38%. We assume that trifluoromethanesulfonic acid not only promotes the main reaction but also catalyzes the side reactions, namely, the hydrolysis of the initial triphenylphosphite **3** and of the target compound **4a**. Furthermore, trifluoromethanesulfonic acid might promote the formation of α-hydroxyphosphonate **6a** (Figure 2). To verify this assumption, we conducted an additional experiment and added acetic anhydride to the reaction mixture with trifluoromethanesulfonic acid as a catalyst in order to chemically bind the water formed during the reaction. Thus, we established that the practical yield of the target compound **4a** sharply increased by up to 75% for a comparable reaction time, which implicitly confirms the assumption (Table 1, entry 6).

If Lewis acids (AlCl_3_, BF_3_·Et_2_O, SnCl_4_ and TiCl_4_) are applied as catalysts in the model reaction to prepare compound **4a** (Table 1, entries 7–9), the yields of the target compound reach 15–25%.

In the course of the further selection of the conditions to prepare compound **4a**, we studied the effect of solvents on the target product yield under the same conditions (room temperature, methanesulfonic acid as a catalyst, 10 mol%). Table 2 shows the results obtained.

Despite a rather low solubility of the initial 1,3,4-trimethylglycoluril **1** in nonpolar solvents at room temperature, the analytical yield of the phosphonate glycoluril **4a** is relatively high and varies from 29 to 59% (Table 2, entries 1–3). At the same time, the maximum analytical yield of 91% (82% being a preparative yield) is achieved in an aprotic medium polar solvent, i.e., acetonitrile. The use of a more polar aprotic solvent, DMSO, does not lead to the formation of the product **4a** in notable amounts according to HPLC. If methanol is applied as a reaction solvent, the highest analytical yield is achieved in 60 min and amounts to only 8%.

It has been experimentally established that an increase in the reaction temperature within 40 to 70 °C—with acetonitrile as a solvent and methanesulfonic acid as a catalyst—led to a decrease in the yield of compound **4a**. Thus, at 40 °C, the maximum analytical yield of product **4a** was 64% for 20 min. At 70 °C, the maximum analytical yield of **4a** equaled 50% for 20 min.

In sum, the optimal conditions for the three-component Birum–Oleksyszyn reaction in the case of 1,3,4-trimethylglycoluril 1 (Figure 1) appear to be the use of acetonitrile as a solvent (room temperature) and the use of methanesulfonic acid as a catalyst (10 mol% of the total reaction mixture). A decrease in the amount of catalyst to 5 mol% or its increase to 20 mol% results in decreases in the yield of the desired product **4a** to 62% and 54%, respectively.

### 2.2. Separation and Identification of Diastereomers of Phosphonate Derivatives of 1,3,4-Trimethylglycoluriles ***4a′*** and ***4a***″

The individual diastereomers **4a′** and **4a**″ were preparatively separated and isolated by HPLC with a C18 stationary phase. The diastereomers were then characterized by NMR and single-crystal X-ray diffraction analysis (SCXRD). Obviously, the order of elution for diastereomers **4a′** and **4a**″ depends on the mutual arrangement of the substituents around the asymmetric C7 centers (Figure 1). The highest retention is characterized by the **4a″** isomer with the most sterically accessible hydrophobic groups.

#### 2.2.1. X-ray Crystal Structure Analysis of the Diastereomers **4a′** and **4a**″

Compound **4a′** crystallizes in a monoclinic crystal system, space group *C2/c*. The asymmetric unit contains one **4a′** molecule and one and a half acetonitrile solvate molecules (Figure 1a). The unit cell contains eight formula units. The crystallographic space group is centrosymmetric; therefore, both enantiomeric forms of compound **4a′** crystallize as a true racemate. The configuration of the stereocenters (3a, 6a, CHPO_3_) in compound **4a′** is as follows: R,S,R (and enantiomeric S,R,S) (Figure 1).

The molecules of **4a′** of the same enantiomeric form are joined into homochiral supramolecular chains by short CH···O contacts involving the oxygen atoms of the phosphonate group and the hydrogen atoms of the phenyl (d(O3–H22) = 2.60 Å) or glycoluril fragments (d(O2–H10) = 2.48 Å, Figure 2a). The second type of CH···O contacts involves the carbonyl oxygen atom and aliphatic hydrogen atoms with the distances d(O4–H7) = 2.43 Å and d(O4–H10) = 2.40 Å, respectively. The chains are oriented along the crystallographic axis b. The enantiomeric chains of opposite stereochemistry are joined in the crystal packing by π-π stacking interactions between the phenyl groups of the phosphonate ester moiety with the distance of 3.295 Å between the planes of the rings (Figure 2b).

Compound **4a**″ crystallizes in a monoclinic centrosymmetric space group, *P2_1_/c*. In contrast to compound **4a′**, there are no solvate molecules in the crystal structure, and the asymmetric unit consists of one molecule (Figure 1b). The unit cell contains four formula units. Similarly to compound **4a′**, the enantiomeric pair of **4a**″ crystallizes as a true racemate. The configuration of the stereocenters (3a, 6a, CHPO_3_) is S,R,R (and R,S,S). Therefore, compounds **4a′** and **4a**″ differ in the absolute configuration of the carbon atom connected to the phosphonate group, in accordance with the two possible directions of the 1,3,4-trimethylglycoluril addition to the carbonyl group of the aldehyde.

Similarly to compound **4a′**, the enantiomers of **4a**″ of the same configuration are joined into supramolecular chains oriented along the crystallographic axis b (Figure 3a), but the short CH···O contacts of the phosphonate groups involve only the hydrogen atoms of the phenyl rings (d(O2–H6) = 2.53 Å and d(O3–H15) = 2.65 Å). The carbonyl oxygen atoms form only one type of short CH···O contacts with the hydrogen atoms of the glycoluril fragment (d(O4–H9) = 2.65 Å). In the crystal packing, the enantiomers of **4a**″ are joined only by C–H···π (d(C19–H25C) = 2.76 Å) contacts, and no other specific supramolecular interactions were found.

#### 2.2.2. Stereoisomerization of Nitro- and Hydroxyphosphonate Derivatives of 1,3,4-Trimethylglycoluriles **4b**–**f**

Further studies were focused on establishing the stereochemical features of obtaining phosphonate derivatives of 1,3,4-trimethylglycoluryl **4b**–**f** with the participation of nitro- and hydroxy derivatives of benzaldehyde **2b**–**f** (Figure 1) using HPLC, NMR spectroscopy, and quantum chemical calculations.

The reactions of nitro- **2b**, **2d**, and **2f** and hydroxybenzaldehydes **2c** and **2e** with 1,3,4-trimethylglycoluril **1** and triphenylphosphite **3** under the selected conditions (acetonitrile and 10 mol% methanesulfonic acid) proceeded with satisfactory yields of **4b**–**4f** (36–67%). According to HPLC, the compounds with structures **4a**″, **4b**″, **4c**″, **4d**″, and **4e**″ were formed in a diastereomeric excess (Figure 1). For compound **4f″**, the diastereomeric excess equaled 96%.

The X-ray crystal structures and NMR results for diastereomers **4a′** and **4a**″ were compared with the NMR spectroscopy data for compounds **4b**–**f**. Thus, for diastereomer **4′**, the signals of the methine protons of ^1^H CH-CH bond were recorded by two doublets within a narrow region of 5.11–5.37 ppm, while for compound **4**″, the signals of these protons were recorded within a broader region of 4.95–5.51 ppm. The ^31^P spectra for the type **4′** diastereomers are characterized by a signal shift towards the weak field relative to the signal of the type **4**″ compounds. The nature of the substituent in the *meta-* and *para-*position does not significantly affect the final ratio of the diastereomers of products **4b**–**e**. However, we observed a notable decrease in the yield of reaction products with a *para-*substituent (the yields for **4d** and **4e** are 40% and 36%, respectively) compared to the reaction products with a *meta-*substituent (yields of **4b** and **4c** are 60% and 67%, respectively). In the case of compound **4f**, only one diastereomer was isolated from the reaction mixture and assigned to the **4f**″ structure. According to HPLC-MS, the application of 2-hydroxybenzaldehyde under the conditions of interest did not allow for the detection of any hydrolysis products of compound **4f′** or any ions with the mass that could be assigned to this compound.

Comparing the positions of diastereomeric structures for the phosphonates of 1,3,4-trimethylglycoluriles (**4a**–**f**) on the energy profile (Appendix A) indicates a larger thermodynamic stability of the **4**″-type structures regardless of the nature and position of the substituent in the aromatic ring. This correlates with the experimental data on the ratio of **4′**:**4**″ stereoisomers (Figure 1). The changes in the electron energy and enthalpy of the isomeric transition of **4**″-type structures to **4′**-type structures range from 2.85 to 3.58 kcal/mol and from 2.92 to 3.93 kcal/mol, respectively (Appendix A).

#### 2.2.3. The Diastereomers of the Phosphonate Derivatives of 1,3,4-Trimethylglycolyrils (**4a**–**f**): General Regularities in the NMR Spectra

NMR spectroscopy was applied to reliably identify the diastereomers of phosphonate derivatives of 1,3,4-trimethylglycolyrils **4a**–**f**. Table 3 presents the results of comparing the experimental and pre-calculated values for the chemical shifts of the hydrogen atoms in bridging the methine groups of the glycoluril fragment (3a and 6a) and the hydrogen atom bonded to the carbon atom in the CHPO_3_ group. The correlations between the experimental and pre-calculated δ values for all of these atoms can be found in Appendix A.

Table 3 shows that the δ for the hydrogen atom bonded to the carbon atom in the CHPO_3_ group, as well as the methine protons of the bridging CH-CH (3a, 6a) group, can indicate the configuration of the phosphonate substituent in structure **4**. Thus, the signals of the protons in the carbon atom with a phosphonate substituent (CHPO_3_) of type **4′** (RSR, SRS) are shifted towards lower values compared to the signals of the enantiomers of type **4**″ (SRR, RSS). For example, the experimental and pre-calculated chemical shifts equal 5.77 and 5.33 ppm, respectively, for the hydrogen atoms bonded to the carbon atom of the phosphonate group (CHPO_3_) in the RSR and SRS enantiomers of compound **4a′**. However, these shifts equal 5.85 and 6.57 ppm, respectively, for the hydrogen atoms of the SRR and RSS enantiomers of compound **4a**″. This pattern persists for the unsubstituted and OH-substituted synthesized glycoluril phosphonates **4a**, **4c**, and **4e**. The experimental signals for similar hydrogen atoms in the indicated pairs of diastereomers of the nitro derivatives (**4b**, **4d**, and **4f**) have identical chemical shifts, while the pre-calculated chemical shifts sustain the trend. At the same time, for δ ^31^P, we observed a significant discrepancy between the pre-calculated values and the experimental ones. This latter fact might be related to the calculation methodology.

A comparison of the experimental and pre-calculated δ values for the hydrogen atoms bonded to the carbon atom in the phosphonate substituent (CHPO_3_) and for the bridging CH-CH (3a, 6a) group demonstrates the significant discrepancy in these values for the enantiomeric pairs of the SRR and RSS (**4**″) types and indicates the values’ precision for the RSR and SRS (**4′**) pairs. For example, the experimental δ of the carbon proton (CHPO_3_) and CH-CH protons (3a, 6a) in the compound of group **4a**″ equal 5.85, 5.50, and 5.00 ppm, whereas the pre-calculated ones equal 6.57, 5.20, and 2.74 ppm. This discrepancy might be caused by the fact that the most stable pre-calculated conformations of structure **4a**″ (SRR and RSS) contain the phenyl rings located relative to the glycoluril fragment, thus creating anisotropy cones and shielding the farthest methine proton in the bridging group (CH-CH). At the same time, the hydrogen atom in the methine group of the phosphonate substituent remains in the de-shielding region (Figure 4). This effect is not observed in the experimental ^1^H NMR spectra, which might be linked to the impossibility of tracking the solvating effect of the solvent using ab initio methods and to the respective change in the structure conformations in the actual solution. The observed effect of phenyl rings on the SRR- and RSS-type enantiomers is retained for the OH and NO_2_ derivatives (**4b**″–**4f**″).

The 2D correlation experiment (NOESY) was performed for compounds **4a′** and **4a**″. The interactions between the protons of the 3a, 6a, and CHPO_3_ sites were identified for compound **4a′**, while these interactions were not detected for compound **4a**″ (Appendix A). 

## 3. Materials and Methods

All reagents (Acros Organics, Merck, Rahway, NJ, USA) were used as purchased, unless indicated otherwise. Solvents for HPLC (PanReac AppliChem, Darmstadt, Germany) were used for the reactions, benzene was dried over molecular sieves 3A for 48 h. 1,3,4-trimethylglycoluril was prepared according to a known procedure [24] and used as a racemate. ^1^H NMR (400 MHz, Chloroform-*d*): δ 7.46 (s, 1H), 5.12 (dd, *J* = 1.7, 8.3 Hz, 1H), 4.97 (d, *J* = 8.3 Hz, 1H), 2.92 (s, 3H), 2.88 (s, 3H), 2.77 (d, *J* = 1.6 Hz, 3H). ^13^C NMR (101 MHz, Chloroform-*d*) δ 160.61, 158.41, 73.83, 65.94, 30.47, 29.55, 27.96.

### 3.1. General Procedure for the Synthesis of Compounds ***4a***–***4f***

Aldehyde **2** (2 mmol), triphenyl phosphite **3** (2 mmol), and methanesulfonic acid (10 mol%) were added to the suspension of 1,3,4-trimethylglycoluril **1** (2 mmol) in dry acetonitrile (4 mL) in an argon atmosphere. The mixture was stirred for 1 h at room temperature and then distilled on a rotary evaporator. The residue was dissolved in 10 mL of toluene and washed with 5% aqueous Na_2_CO_3_ solution (3 × 10 mL). The toluene solution was then washed with water (3 × 10 mL). Upon washing, the organic layer was dried over Na_2_SO_4_ and concentrated on a rotary evaporator. Concentrated viscous oil was purified by preparative HPLC with C18 stationary phase and water–acetonitrile (60:40) mobile phase.

### 3.2. NMR Spectroscopy

^1^H, ^13^C, and ^31^P NMR spectra were recorded using the Bruker AVANCE III HD 400 MHz NMR spectrometer (Brucker BioSpin GmbH, Ettlingen, Germany) with the PA BBO 400S1 BBF-H-D-05 Z SP probe head and the BCU temperature control unit, PLC on TTY1 of ELCB 1 autosampler, and TopSpin 3.5 pl5 interface. The ^1^H and ^13^C NMR signals of the samples were assigned to the ^1^H and ^13^C NMR signals of tetramethylsilane (TMS) (0.0 ppm), and the ^31^P NMR signals were assigned to the ^31^P NMR signals of H_3_PO_4_ (0.0 ppm). Appendix A show the ^1^H, ^13^C, and ^31^P NMR spectra of the compounds.

*Diphenyl ((1S)-phenyl(3,4,6-trimethyl-2,5-dioxohexahydroimidazo[4,5-d]imidazol-1(2H)-yl)methyl)phosphonate* (**4a′**), white solid. ^1^H NMR (400 MHz, DMSO-*d*_6_): δ 7.75–7.70 (m, 2H), 7.54–7.45 (m, 4H), 7.42–7.33 (m, 4H), 7.28–7.19 (m, 2H), 7.12–7.05 (m, 2H), 6.97 (dt, *J* = 8.5, 1.2 Hz, 2H), 5.77 (d, *J* = 27.4 Hz, 1H), 5.28 (dd, *J* = 8.5, 1.3 Hz, 1H), 5.24 (d, *J* = 8.5 Hz, 1H), 2.94 (s, 3H), 2.88 (s, 3H), 2.57 (s, 3H). ^13^C NMR (101 MHz, DMSO): 159.06, 158.37 (d, *J* = 3.7 Hz), 150.59 (d, *J* = 9.9 Hz), 134.76, 130.17 (d, *J* = 18.7 Hz), 129.45 (d, *J* = 6.6 Hz), 129.12, 125.55 (d, *J* = 15.0 Hz), 120.90 (dd, *J* = 9.1, 4.1 Hz), 72.40, 72.14 (d, *J* = 5.0 Hz), 57.37 (d, *J* = 155.9 Hz), 30.70, 30.27, 30.13; ^31^P NMR (162 MHz, DMSO-d6): δ 14.06 (d, *J* = 27.1 Hz). MS (HRMS-ESI): Calcd. For C_26_H_27_N_4_O_5_P, [M + H]^+^: 507.1792, found: *m/z* 507.1823.

*Diphenyl ((1R)-phenyl(3,4,6-trimethyl-2,5-dioxohexahydroimidazo[4,5-d]imidazol-1(2H)-yl)methyl)phosphonate* (**4a**″) white solid. ^1^H NMR (400 MHz, DMSO-*d*_6_): δ 7.65 (d, *J* = 7.5 Hz, 2H), 7.50–7.35 (m, 8H), 7.30–7.21 (m, 2H), 7.17 (d, *J* = 8.1 Hz, 2H), 7.13–7.07 (m, 2H), 5.85 (d, *J* = 25.8 Hz, 1H), 5.50 (d, *J* = 8.5 Hz, 1H), 5.00 (d, *J* = 8.5 Hz, 1H), 2.88 (s, 3H), 2.82 (s, 3H), 2.16 (s, 3H). ^13^C NMR (101 MHz, DMSO): δ 159.41, 158.11 (d, *J* = 2.9 Hz), 150.02 (dd, *J* = 9.7, 3.3 Hz), 130.57 (d, *J* = 7.9 Hz), 129.28, 129.03, 129.00, 128.95, 126.15, 120.73 (dd, *J* = 20.8, 4.1 Hz), 72.32, 70.78, 53.84 (d, *J* = 155.3 Hz), 30.85, 30.75, 30.26; ^31^P NMR (162 MHz, DMSO-*d*_6_): δ 13.35 (d, *J* = 25.9 Hz). MS (HRMS-ESI): Calcd. for C_26_H_27_N_4_O_5_P, [M + H]^+^: 507.1792, found: *m*/*z* 507.1825.

*Diphenyl ((1S)-(3-nitrophenyl)(3,4,6-trimethyl-2,5-dioxohexahydroimidazo[4,5-d]imidazol-1(2H)-yl)methyl)phosphonate* (**4b′**) yellow viscous oil. ^1^H NMR (400 MHz, DMSO-*d*_6_): δ 8.57 (s, 1H), 8.28–8.24 (m, 1H), 8.08 (d, *J* = 5.7 Hz, 1H), 7.77 (t, *J* = 8.0 Hz, 1H), 7.46–7.33 (m, 4H), 7.32–7.24 (m, 2H), 7.24–7.16 (m, 2H), 7.14–7.07 (m, 2H), 5.98 (d, *J* = 27.4 Hz, 1H), 5.36 (d, *J* = 8.4 Hz, 1H), 5.25 (d, *J* = 8.5 Hz, 1H), 2.89 (s, 3H), 2.85 (s, 3H), 2.49 (s, 3H). ^13^C NMR (101 MHz, DMSO-*d*_6_): δ 158.79, 158.42 (d, *J* = 3.1 Hz), 150.59 (d, *J* = 10.2 Hz), 150.45 (d, *J* = 10.0 Hz), 148.22, 137.96 (d, *J* = 2.3 Hz), 135.77 (d, *J* = 5.9 Hz), 130.72, 130.39, 130.08, 125.79, 125.50, 124.10, 124.04, 123.72, 120.82, 120.78, 120.72, 120.68, 72.69 (d, *J* = 4.9 Hz), 72.41, 57.44 (d, *J* = 152.4 Hz), 30.75, 30.16, 29.86. ^31^P NMR (162 MHz, DMSO-*d*_6_): δ 12.77 (d, *J* = 27.8 Hz); MS (HRMS-ESI): Calcd. for C_26_H_26_N_5_O_7_P, [M + H]^+^: 552.1643, found: *m*/*z* 552.1641.

*Diphenyl ((1R)-(3-nitrophenyl)(3,4,6-trimethyl-2,5-dioxohexahydroimidazo[4,5-d]imidazol-1(2H)-yl)methyl)phosphonate* (**4b″**) yellow viscous oil. 1H NMR (400 MHz, DMSO-*d*_6_): δ 8.57 (s, 1H), 8.28 (dd, *J* = 2.3, 8.1 Hz, 1H), 8.10 (d, *J* = 7.7 Hz, 1H), 7.78 (t, *J* = 8.1 Hz, 1H), 7.45–7.36 (m, 5H), 7.31–7.24 (m, 2H), 7.20–7.16 (m, 2H), 7.13–7.09 (m, 2H), 5.98 (d, *J* = 27.0 Hz, 1H), 5.51 (d, *J* = 8.5 Hz, 1H), 5.10 (d, *J* = 8.5 Hz, 1H), 2.90 (s, 3H), 2.85 (s, 3H), 2.40 (s, 3H).^13^C NMR (101 MHz, DMSO-*d*_6_): δ 159.47, 157.94 (d, *J* = 2.9 Hz), 150.00 (d, *J* = 5.5 Hz), 149.90 (d, *J* = 5.4 Hz), 148.17, 136.84 (d, *J* = 6.4 Hz), 135.68 (d, *J* = 7.9 Hz), 130.89, 130.65, 130.52, 126.30, 126.18, 123.95, 123.71, 123.63, 120.83, 120.78, 120.72, 120.68, 72.46, 71.00, 54.02 (d, *J* = 157.3 Hz), 31.08, 30.86, 30.29. ^31^P NMR (162 MHz, DMSO-*d*_6_): δ 12.32 (d, *J* = 26.8 Hz); MS (HRMS-ESI): Calcd. for C_26_H_26_N_5_O_7_P, [M + H]^+^: 552.1643, found: *m*/*z* 552.1646.

*Diphenyl ((1S)-(3-hydroxyphenyl)(3,4,6-trimethyl-2,5-dioxohexahydroimidazo[4,5-d]imidazol-1(2H)-yl)methyl)phosphonate* (**4c′**) white viscous oil. ^1^H(400 MHz, DMSO), δ, ppm: 9.63 (s, 1H), 7.38–7.29 (m, 4H), 7.26–7.12 (m, 6H), 7.04 (d, *J* = 8.1 Hz, 2H), 6.99–6.91 (m, 2H), 6.78 (d, *J* = 8.2 Hz, 1H), 5.64 (d, *J* = 27.2 Hz, 1H), 5.15 (s, 2H), 2.88 (s, 3H), 2.82 (s, 3H), 2.56 (s, 3H). ^13^C NMR (101 MHz, DMSO), δ, ppm: 157.04, 156.22 (d, *J* = 3.7 Hz), 155.82, 128.03 (d, *J* = 13.3 Hz), 123.42 (d, *J* = 7.3 Hz), 118.81 (dd, *J* = 16.8, 4.1 Hz), 118.48, 117.94, 114.21, 113.79, 70.30, 69.79, 54.92 (d, *J* = 158.1 Hz), 28.57, 28.17, 28.14; ^31^P (162 MHz, DMSO), δ, ppm: 14.16 (d, *J* = 27.1 Hz). MS (HRMS-ESI): Calcd. for C_26_H_27_N_4_O_6_P, [M + H]^+^: 523.1741, found: *m*/*z* 523.1777.

*Diphenyl ((1R)-(3-hydroxyphenyl)(3,4,6-trimethyl-2,5-dioxohexahydroimidazo[4,5-d]imidazol-1(2H)-yl)methyl)phosphonate* (**4c**″) white viscous oil. ^1^H(400 MHz, DMSO), δ, ppm: δ 9.67 (s, 1H), 7.54–7.34 (m, 4H), 7.30–7.20 (m, 3H), 7.20–7.15 (m, 2H), 7.15–7.04 (m, 4H), 6.78 (dd, *J* = 2.4, 8.1 Hz, 1H), 5.76 (d, *J* = 25.9 Hz, 1H), 5.49 (d, *J* = 8.5 Hz, 1H), 4.97 (d, *J* = 8.6 Hz, 1H), 2.88 (s, 3H), 2.82 (s, 3H), 2.19 (s, 3H). ^13^C NMR (101 MHz, DMSO), δ, ppm: 157.27, 155.99 (d, *J* = 3.1 Hz), 155.90, 147.92 (dd, *J* = 9.7, 6.4 Hz), 133.51 (d, *J* = 7.3 Hz), 128.43 (d, *J* = 6.5 Hz), 124.00, 118.59 (dd, *J* = 25.6, 4.1 Hz), 117.32 (d, *J* = 8.1 Hz), 113.86, 113.73 (d, *J* = 8.8 Hz), 70.16, 68.62, 51.47 (d, *J* = 155.0 Hz), 28.62, 28.12; ^31^P (162 MHz, DMSO), δ, ppm: 13.35 (d, *J* = 25.7 Hz); MS (HRMS-ESI): Calcd. for C_26_H_27_N_4_O_6_P, [M + H]^+^: 523.1741, found: *m*/*z* 523.1780.

*Diphenyl ((1S)-(4-nitrophenyl)(3,4,6-trimethyl-2,5-dioxohexahydroimidazo[4,5-d]imidazol-1(2H)-yl)methyl)phosphonate* (**4d′**) yellow viscous oil. ^1^H (400 MHz, Chloroform-*d*), δ, ppm: 8.19 (2H, d, *J* = 8.1 Hz), 7.84 (2H, d, *J* = 8.2 Hz), 7.35–7.20 (4H, m), 7.21–7.02 (6H, m), 5.26 (1H, d, *J* = 26.4 Hz), 4.93 (2H, d, *J* = 5.1 Hz), 2.95 (3H, s), 2.88 (3H, s), 2.83 (3H, s); ^13^C NMR (101 MHz, Chloroform-*d*), δ, ppm: 158.42, 158.22 (d, *J* = 4.2 Hz), 150.15 (d, *J* = 9.7 Hz), 147.94 (d, *J* = 3.1 Hz), 140.66, 130.19, 129.96 (d, *J* = 6.0 Hz), 129.69 (d, *J* = 8.7 Hz), 125.97 (d, *J* = 28.7 Hz), 125.55 (d, *J* = 24.6 Hz), 123.97 (d, *J* = 14.4 Hz), 120.42, 120.26 (d, *J* = 4.3 Hz), 119.99 (d, *J* = 4.0 Hz), 72.77, 72.26, 57.88 (d, *J* = 156.2 Hz), 30.65, 30.49, 30.04; ^31^P (162 MHz, Chloroform-*d*), δ, ppm: 11.29 (d, *J* = 26.4 Hz); MS (HRMS-ESI): Calcd. for C_26_H_26_N_5_O_7_P, [M + H]^+^: 552.1643, found: *m*/*z* 552.1674.

*Diphenyl ((1R)-(4-nitrophenyl)(3,4,6-trimethyl-2,5-dioxohexahydroimidazo[4,5-d]imidazol-1(2H)-yl)methyl)phosphonate* (**4d**″) yellow viscous oil. ^1^H (400 MHz, Chloroform-*d*), δ, ppm: 8.15 (2H, d, *J* = 8.2 Hz), 7.93 (2H, d, *J* = 7.9 Hz), 7.34–7.24 (4H, m), 7.20–7.10 (6H, m), 6.18 (1H, d, *J* = 26.2 Hz), 5.43 (1H, d, *J* = 8.6 Hz), 4.29 (1H, dd, *J* = 8.6, 1.4 Hz), 2.90 (2H, s), 2.82 (2H, s), 2.09 (2H, s); ^13^C NMR (101 MHz, Chloroform-*d*), δ, ppm: 157.03, 155.93 (d, *J* = 4.1 Hz), 148.27 (d, *J* = 9.0 Hz), 147.43 (d, *J* = 10.6 Hz), 146.30, 134.72 (d, *J* = 8.8 Hz), 132.98 (d, *J* = 9.2 Hz), 128.22, 128.01, 124.17, 123.84, 121.68 (d, *J* = 7.1 Hz), 118.46 (d, *J* = 4.2 Hz), 118.06 (d, *J* = 4.4 Hz), 70.31, 68.62, 50.53 (d, *J* = 158.2 Hz), 29.22, 28.69, 28.57; ^31^P (162 MHz, Chloroform-*d*), δ, ppm: 10.48 (d, *J* = 26.2 Hz); MS (HRMS-ESI): Calcd. for C_26_H_26_N_5_O_7_P, [M + H]^+^: 552.1643, found: *m*/*z* 552.1670.

*Diphenyl ((1S)-(4-hydroxyphenyl)(3,4,6-trimethyl-2,5-dioxohexahydroimidazo[4,5-d]imidazol-1(2H)-yl)methyl)phosphonate* (**4e′**) white viscous oil. ^1^H(400 MHz, DMSO), δ, ppm: 9.72 (s, 1H), 7.50 (d, *J* = 7.8 Hz, 2H), 7.39–7.27 (m, 5H), 7.20–7.14 (m, 3H), 7.10–7.04 (m, 2H), 6.98–6.92 (m, 2H), 6.84 (d, *J* = 7.6 Hz, 2H), 5.63 (d, *J* = 26.5 Hz, 1H), 5.11 (s, 2H), 2.88 (s, 3H), 2.81 (s, 3H), 2.63 (s, 3H). ^13^C NMR (101 MHz, DMSO), δ, ppm: 157.19, 156.37 (d, *J* = 4.3 Hz), 155.69, 148.53 (t, *J* = 10.0 Hz), 129.17 (d, *J* = 7.3 Hz), 128.05 (d, *J* = 11.1 Hz), 123.42 (d, *J* = 5.5 Hz), 118.89 (d, *J* = 4.0 Hz), 118.74 (d, *J* = 4.0 Hz), 113.88, 70.40, 69.61, 54.34 (d, *J* = 160.3 Hz), 29.03, 28.55, 28.20; ^31^P (162 MHz, DMSO), δ, ppm: 14.56 (d, *J* = 26.3 Hz); MS (HRMS-ESI): Calcd. for C_26_H_27_N_4_O_6_P, [M + H]^+^: 523.1741, found: *m*/*z* 523.1767.

*Diphenyl ((1R)-(4-hydroxyphenyl)(3,4,6-trimethyl-2,5-dioxohexahydroimidazo[4,5-d]imidazol-1(2H)-yl)methyl)phosphonate* (**4e**″) white viscous oil. ^1^H(400 MHz, DMSO), 9.70 (1H, s), 7.50 (2H, d, *J* = 7.8 Hz) 7.47–7.34 (5H, m), 7.24 (3H, dt, *J* = 14.5, 7.4 Hz), 7.16 (2H, d, *J* = 8.0 Hz), 7.08 (2H, d, *J* = 8.0 Hz), 6.81 (2H, d, *J* = 8.5 Hz), 5.71 (1H, d, *J* = 25.2 Hz), 5.48 (1H, d, *J* = 8.4 Hz), 4.95 (1H, d, *J* = 8.4 Hz), 2.85 (3H, s), 2.81 (3H, s), 2.16 (3H, s); ^13^C NMR (101 MHz, DMSO), δ, ppm: 159.34, 158.13 (d, *J* = 2.3 Hz), 158.02, 150.10 (dd, *J* = 9.9, 4.6 Hz), 130.71 (d, *J* = 8.6 Hz), 130.53 (d, *J* = 8.4 Hz), 126.06, 120.86 (d, *J* = 4.0 Hz), 120.60 (d, *J* = 4.1 Hz), 115.99, 72.27, 70.64, 53.36 (d, *J* = 155.0 Hz), 30.82, 30.64, 30.19; ^31^P (162 MHz, DMSO), δ, ppm:13.87 (d, *J* = 25.4 Hz); MS (HRMS-ESI): Calcd. for C_26_H_27_N_4_O_6_P, [M + H]^+^: 523.1741, found: *m*/*z* 523.1776.

*Diphenyl ((1S)-(2-nitrophenyl)(3,4,6-trimethyl-2,5-dioxohexahydroimidazo[4,5-d]imidazol-1(2H)-yl)methyl)phosphonate* (**4f**″) yellow viscous oil. ^1^H(400 MHz, DMSO), δ, ppm: 8.24 (d, *J* = 8.0 Hz, 1H), 8.05 (d, *J* = 8.1 Hz, 1H), 7.86 (td, *J* = 1.4, 7.8 Hz, 1H), 7.73 (t, *J* = 7.7 Hz, 1H), 7.44–7.17 (m, 7H), 7.17–7.03 (m, 2H), 7.01–6.94 (m, 2H), 6.35 (d, *J* = 26.2 Hz, 1H), 5.49 (d, *J* = 8.4 Hz, 1H), 5.11 (d, *J* = 8.4 Hz, 1H), 2.84 (d, *J* = 5.5 Hz, 6H), 2.39 (s, 3H). ^13^C NMR (101 MHz, DMSO), δ, ppm: 159.46, 157.74, 150.21–149.95 (m), 133.92, 132.08 (d, *J* = 4.3 Hz), 131.19, 130.47 (d, *J* = 17.6 Hz), 127.62 (d, *J* = 7.0 Hz), 126.06 (d, *J* = 9.2 Hz), 125.74, 120.56 (dd, *J* = 21.0, 4.2 Hz), 72.48, 70.83 (d, *J* = 3.2 Hz), 50.04 (d, *J* = 158.9 Hz), 30.90, 30.76, 30.29; ^31^P (162 MHz, DMSO), δ, ppm: 11.67 (d, *J* = 26.4 Hz); MS (HRMS-ESI): Calcd. for C_26_H_26_N_5_O_7_P, [M + H]^+^: 552.1643, found: *m*/*z* 552.1692.

### 3.3. X-ray Crystallography

The X-ray diffraction data were collected at 150 °K with the Bruker D8 Venture diffractometer (Bruker Optik GmbH, Ettlingen, Germany) (0.5° ω- and φ-scans, fixed-χ three-circle goniometer, CMOS PHOTON III detector, Mo-IμS 3.0 microfocus source, focusing Montel mirrors, λ = 0.71073 Å MoK_α_ radiation, N_2_-flow thermostat). Data reduction was performed via the APEX 3 suite. The crystal structure was solved using the ShelXT [25] and was refined using the ShelXL [26] programs assisted by the Olex2 GUI [27]. The atomic displacements for the non-hydrogen atoms were refined with harmonic anisotropic approximation. Hydrogen atoms were located geometrically and refined in the riding model. Table 4 represents the crystallographic parameters for compounds **4a′** and **4a″**.

### 3.4. HPLC Analysis

The analysis was performed with the Agilent 1200 chromatograph (Agilent Technologies, Santa Clara, CA, USA). Spectrophotometric detection was carried out using the Zorbax Eclipse Plus C18 4.6 × 100 mm 3.5 μm chromatographic column (Agilent Technologies, Santa Clara, CA, USA) as a stationary phase. The chromatographic mode was gradient, mobile phase A was a buffer solution (25 mM formic acid (Honeywell Fluka, Charlotte, NC, USA) and 5 mM ammonia (Honeywell Fluka, Charlotte, NC, USA)), and mobile phase B was acetonitrile (PanReac AppliChem, Darmstadt, Germany). Gradient mode: 0.0 min–50% A, 0.5 min–50% A, 7.5 min–20% A, 9.0–20% A, 9.01–50% A, 11.0–50% A. Flow rate was 1.5 mL/min, and thermostat temperature was 25 °C.

### 3.5. Preparative HPLC

The preparative purification and separation of diastereomers **4a′**–**e′** and **4a**″–**f**″ were performed using the Shimadzu LC 20 Prominence chromatograph (Shimadzu, Kyoto, Japan) with the Kromasil C18 column 20 × 250 mm, 5 µm particle size (Nouryon, Bohus, Sweden). The column temperature was set to 25 °C (±1 °C). The mobile phase consisted of a water–acetonitrile mixture (60:40). The flow rate was 5 mL/min in the isocratic mode. The detection wavelength for the UV detector equaled 250 nm. The samples were preliminarily dissolved in acetonitrile (1:10, *v*/*v*). Appendix A show the HRMS of the compounds **4a**–**4f**.

### 3.6. HRMS

The analysis was performed using the HPLC-MS method in the electro-spray ionization mode using the Agilent QTOF-6550 (Agilent Technologies, Santa Clara, CA, USA) connected with Agilent 1260 Infinity II (Agilent Technologies, Santa Clara, CA, USA). The samples were analyzed with the Poroshell 120 EC-C18 column 2.1 × 100 mm, 2.7 µm particle size. The column temperature was set as 30 °C (±0.4 °C). The chromatographic analysis was performed in isocratic mode. The 0.05% formic acid (Honeywell Fluka, Charlotte, NC, USA) aqueous solution–acetonitrile mixture (60:40) was applied as a mobile phase. The flow rate was 0.5 mL/min. The samples were analyzed in a positive ionization mode with a scan range of 300–900 Da. Capillary and nozzle voltages were set to 2000 and 100 V, respectively. Drying gas temperature was 200 °C, and sheath gas temperature was 250 °C. The drying gas flow was 13 L/min, and the sheath gas flow was 11 L/min. The samples were preliminarily dissolved in acetonitrile (1:1000, *v*/*v*). Appendix A show the preparative elution profiles of the compounds **4a**–**4f**.

### 3.7. Computational Details

All model calculations were performed using the Gaussian’09 program package [28] installed on the SKIF “Cyberia” supercomputer of Tomsk State University. The geometries of the structures under analysis were fully optimized at the meta-hybrid functional M062X [29] with the split-valence basis set 6-311+G(2d,p) in DMSO (ε = 46.826) as a solvent using PCM. The PCM was applied using a scaled van der Waals surface cavity with an alpha value of 1.1, and atomic radii modelling using a universal forcefield. The geometry optimization was performed on the basis of the experimentally determined X-ray structures of **4a’** and **4a**″ without conformational analysis. Frequency calculations were also performed at the M062X/6-311+G(2d,p) level of theory. The freely available GoodVibes script developed by Paton and Funes-Ardoiz [30] was used to recalculate the thermochemical data with the low-frequency corrections and correction from the gas with the pressure of 1 atm to solution with the concentration of 1 mol/L.

Appendix A show the correlation between the geometric parameters for the optimized **4a** molecular structures and the experimental data. Appendix A presents the XYZ coordinates for all of the compounds under analysis.

Magnetic shielding tensors were calculated with the gauge including atomic orbitals (GIAO) DFT method using the Gaussian’09 at the PBE0/6-311+G(2d,p) level theory.

The chemical shifts (δ ^31^Pcalc) were detected as δ ^31^Pcalc = σ H_3_PO_4_–σ calc, where σ H_3_PO_4_ is the shielding constant of ^31^P in phosphoric acid (H_3_PO_4_). The σ H_3_PO_4_ values of ^31^P of the H_3_PO_4_ were computed using the same strategy as for structure **4**. Appendix A shows the correlations between experimental and calculation values δ ^1^H, ^13^C atoms of 3a, 6a, CHPO_3_ and δ ^31^P of the compounds **4a–4f**.

## 4. Conclusions

The results demonstrate an approach to the synthesis of new phosphonate derivatives of 1,3,4-trimethylglycoluril via the Birum–Oleksyszyn reaction. This approach allowed for the obtainment of a series of diastereomers that were formed in the reaction medium with the predominance of structure **4″**. The obtained diastereomers can be separated by the preparative HPLC with the C18 stationary phase. The diastereomers’ structures were confirmed via NMR spectroscopy and HRMS; for compound **4a**, the structure was confirmed with a single-crystal X-ray diffraction analysis. The results of the quantum chemical calculations were consistent with the experimental data. Additionally, the authors revealed a correlation between the spatial arrangement of substituents in the structure of 1-[aryl-(diphenylphosphono)methyl]-3,4,6-trimethylglycolurils and the δ values of the proton on the carbon atom in the phosphonate group (CHPO_3_) as well as the proton bridging CH-CH group. For the compounds with structure **4′**, the signals of the methine protons in the CH-CH bond were recorded in the ^1^H NMR spectrum as two doublets within the narrow region (5.11–5.37 ppm), while for the compounds with structure **4″,** the signals of these protons were recorded within a wider range (4.95–5.51 ppm). At the same time, the signal of the proton bonded to the carbon atom in the phosphonate group (CHPO_3_) was shifted to a higher field for the compounds with structure **4′** compared to the similar proton in structure **4**″.

## Data Availability

The data that support the findings of this study are available within the article and the Appendix A. Further data are available from the corresponding author upon reasonable request. Crystallographic Data Centre, CCDC no. 2294213 provided compound **4a’**, and CCDC no. 2294214 provided compound **4a**″. Copies of the data can be obtained free of charge from the Cambridge Crystallographic Data Centre, 12 Union Road, Cambridge CB2 1EZ, UK (fax: +44-1223-336-033; e-mail: deposit@ccdc.cam.ac.uk).

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
