# Peer review of "Synthesis of Novel Phosphorus-Containing Derivatives of 1,3,4-Trimethylglycoluril via the Birum–Oleksyszyn Reaction"

_ijms, 2023, doi:10.3390/ijms242317082_

Round 1

Reviewer 1 Report

Comments and Suggestions for Authors

In the present manuscript, the authors reported the synthesis of novel phosphorous-containing derivatives of 1,3,4- trimethylglycolurile by Birum-Oleksyszyn reaction starting from aromatic aldehydes. The authors proposed the reaction as a methodology for achieving 1,3,4  trimethylglycoluryl compounds derivated by aliphatic aldehydes in a recent manuscript (rif 18) in which they reported the chemical conditions (ACN,  methanesulfonic acid, room temperature)  for the synthesis with a good yield of desired compounds. In the present manuscript, they evaluated different reaction conditions and catalysts for the preparation of 4a compound obtained with 91% yield by using the same conditions proposed in the previous publication. The question is: why did not the authors directly use these chemical conditions and spend the time to search for others?  Where is the novelty of the manuscript about the synthetic part?

Comments

·         INTRODUCTION

Do phosphorus-containing glycoluril derivatives have only catalytic activity? The authors could emphasize other potential activities of the compounds.

Lanes 30-32: in the sentence “…these reports are sporadic and to date have developed a relatively small number of synthetic approaches for the preparation of phosphorus-containing glycoluril derivatives”,  the reference(s) should be inserted.

·         RESULTS

The authors should include the numbering of the atoms in the structure 4 a-f in scheme 1 to make the manuscript easier to read.

In the text, the trifluoromethanesulfonic acid is also reported as triflic acid. Ti should be better to replace triflic with trifluoromethanesulfonic acid.

Lane 107. In the sentence: “To verify this assumption, we conducted an experiment in which, using trifluoromethane acid as a catalyst….” Does trifluoromethane acid mean trifluoromethanesulfonic acid?

Lane 123. In the sentence….” analytical yield of 4a 91 % (preparative 81 %)….” the preparative yield is not in accordance with the value reported in Table 1 (82%).

Lanes 134. “Olexisne reaction” must be replaced with “Oleksyszyn reaction”

Lane 154: Scheme 1 or Figure 1?

Lane 153. Are the configurations of the stereocenters reported in Table 3? Could the authors check the stereochemistry stated in the text and in the table, please?

Lanes 200-201. The sentence “According to HPLC analysis, compounds with structures 4a″, 4b″, 4c″, 4d″, and 200 4e″ are formed in a small diastereomeric excess during the reaction (Scheme 3)” is not in accordance with the ratio of diastereomers reported in the scheme 3 where 4a”-4e” excess is evident.

Lane 231. The fragment sentence “glycoluril fragment (10C and 12C) and hydrogen atom at chiral carbon atom (5C) “must

be replaced with “glycoluril fragment (10C and 5C) and hydrogen atom at chiral carbon atom (12C).

·         MATERIALS AND METHODS

Lane 422. “HPLC analisys” have to be replaced with “HPLC analysis”

Regarding the analytical and preparative methods why do the authors use different elution modes? Gradient mode and isocratic one?

·         REFERENCES

The authors must check the format of the reference 17.

·         CONCLUSIONS

Lane 472. What the authors mean by “C18 stratigraphic phase”

Could the different spatial arrangement of substituents in 4’ and 4’’ derivatives correlate with the yield % of 4 in the different solvents? (table2). The authors should try to find any correlations 

Comments on the Quality of English Language

Minor editing of English language required

Author Response

Dear Reviewer,

Thank you for your attention to our manuscript. This careful review helped us to better understand the results obtained. All comments and recommendations helped us to improve the manuscript. The comments and corrections to the review are presented below.

Reviewer comment

In the present manuscript, the authors reported the synthesis of novel phosphorous-containing derivatives of 1,3,4- trimethylglycolurile by Birum-Oleksyszyn reaction starting from aromatic aldehydes. The authors proposed the reaction as a methodology for achieving 1,3,4  trimethylglycoluryl compounds derivated by aliphatic aldehydes in a recent manuscript (rif 18) in which they reported the chemical conditions (ACN,  methanesulfonic acid, room temperature)  for the synthesis with a good yield of desired compounds. In the present manuscript, they evaluated different reaction conditions and catalysts for the preparation of 4a compound obtained with 91% yield by using the same conditions proposed in the previous publication. The question is: why did not the authors directly use these chemical conditions and spend the time to search for others?  Where is the novelty of the manuscript about the synthetic part?

Author response

The results presented in this paper are a part of a general work on the preparation of phosphonate derivatives of methylglycoluryl compounds. The work on the preparation of 1,3,4-trimethylglycoluryl with aliphatic aldehydes was published as a short communication, which described only the synthesis results. However, the search for the synthesis conditions to achieve satisfactory and good yields of the phosphonate derivatives of 1,3,4-trimethylglycoluryl compounds was not shown in the paper (Ref. 18 (the reference number was changed to Ref. 20)). In our opinion, the data on the search for the reaction conditions (the influence of the nature of the catalyst, solvent as well as time) presented in this paper can be useful for the chemistry of glycolurils, namely, can be extended to the preparation of new derivatives of phosphonate glycolurils.

Reviewer comment

Introduction

Do phosphorus-containing glycoluril derivatives have only catalytic activity? The authors could emphasize other potential activities of the compounds.

Author response

Thank you for your comment. The literature also describes the phosphorus-containing glycolurils capable of imparting fire-resistant properties to rubbers [DOI:10.1134/S1070427216010213]. Ref. [DOI:10.1002/chem.201902556] also describes the molecular tweezer that exhibits high binding properties to aliphatic amines. In view of the fact that some alkyl derivatives of glycoluryl and aminophosphonates exhibit biological activity, the glycolurils described in this work may have potential biological activity. The above formulations were added to the introduction section.

Reviewer comment

Lanes 30-32: in the sentence “…these reports are sporadic and to date have developed a relatively small number of synthetic approaches for the preparation of phosphorus-containing glycoluril derivatives”,  the reference(s) should be inserted.

Author response

Thank you for your comment. Additional references to sources 18 and 19 were added (Lines 30-31).

Reviewer comment

RESULTS

The authors should include the numbering of the atoms in the structure 4 a-f in scheme 1 to make the manuscript easier to read.

Author response

Thank you for your comment. The numbering of some atoms was included in Scheme 1.

Reviewer comment

In the text, the trifluoromethanesulfonic acid is also reported as triflic acid. Ti should be better to replace triflic with trifluoromethanesulfonic acid.

Author response

Thank you for your comment. The text was revised accordingly.

Reviewer comment

Lane 107. In the sentence: “To verify this assumption, we conducted an experiment in which, using trifluoromethane acid as a catalyst….” Does trifluoromethane acid mean trifluoromethanesulfonic acid?

Author response

Thank you for your comment. The sentence was revised accordingly.

Reviewer comment

Lane 123. In the sentence….” analytical yield of 4a 91 % (preparative 81 %)….” the preparative yield is not in accordance with the value reported in Table 1 (82%).

Author response

Thank you for your comment. The sentence was revised accordingly.

Reviewer comment

Lanes 134. “Olexisne reaction” must be replaced with “Oleksyszyn reaction”

Author response

Thank you for your comment. The sentence was revised accordingly.

Reviewer comment

Lane 154: Scheme 1 or Figure 1?

Author response

Thank you for your comment. “Fig 1” has been clarified on line 150.

Reviewer comment

Lane 153. Are the configurations of the stereocenters reported in Table 3? Could the authors check the stereochemistry stated in the text and in the table, please?

Author response

Thank you for your comment. We checked the stereocentre configuration in Table 3 as well as in section 2.2.3.

Reviewer comment

Lanes 200-201. The sentence “According to HPLC analysis, compounds with structures 4a″, 4b″, 4c″, 4d″, and 200 4e″ are formed in a small diastereomeric excess during the reaction (Scheme 3)” is not in accordance with the ratio of diastereomers reported in the scheme 3 where 4a”-4e” excess is evident.

Author response

Thank you for your comment. The ambiguous characterization of the diastereomeric excess was removed.

Reviewer comment

Lane 231. The fragment sentence “glycoluril fragment (10C and 12C) and hydrogen atom at chiral carbon atom (5C) “must be replaced with “glycoluril fragment (10C and 5C) and hydrogen atom at chiral carbon atom (12C).

Author response

Thank you for your comment. The sentence was revised accordingly.

  • MATERIALS AND METHODS

Reviewer comment

Lane 422. “HPLC analisys” have to be replaced with “HPLC analysis”

Author response

Thank you for your comment. The text was revised accordingly.

Reviewer comment

Regarding the analytical and preparative methods why do the authors use different elution modes? Gradient mode and isocratic one?

Author response

Thank you for your comment. The preparative separation was carried out in the isocratic mode due to the absence of a gradient mixer in the preparative chromatograph.

Reviewer comment

REFERENCES

The authors must check the format of the reference 17.

Author response

Thank you for your comment. The reference format was revised.

Reviewer comment

CONCLUSIONS

Lane 472. What the authors mean by “C18 stratigraphic phase”

Author response

Thank you for your comment. This is referred to the stationary phase. The text was revised.

Reviewer comment

Could the different spatial arrangement of substituents in 4’ and 4’’ derivatives correlate with the yield % of 4 in the different solvents? (table2). The authors should try to find any correlations 

Author response

Thank you for the comment. We analyzed the isomeric composition of the products in different solvents. No unambiguous correlations of stereoisomer ratios with dielectric constant were found.

Reviewer comment

Comments on the Quality of English Language

Minor editing of English language required

Author response

Thank you for the comment. English was proof-read by a native speaker.

Sincerely yours and on behalf of the authors,

Vera P. Tuguldurova

Reviewer 2 Report

Comments and Suggestions for Authors

The present manuscript describes the synthesis of several (diphenylphosphono)methyl derivatives of 1,3,4-trimethylglycolurile and optimal conditions for the highest possible yields of the target products are presented. The final products are separated by HPLC and characterization by XRD and NMR with respect to stereochemistry.

Major issues

  1. The diastereomers 4a’ and 4a’’ were separated by HPLC and their structures were obtained from single-crystal XRD. This makes a solid ground for unambiguous interpretation of the NMR spectra. However, 4b-f were not analyzed by XRD.

    1. How can the authors prove that the 4’ vs. 4’’ stereoisomers of 4b-f are the ones that are presented in the manuscript? Is the order and separation in HPLC clear enough? The elusion profiles should be part of the publication.

    2. Solution conformation is not discussed. One can expect several exchanging geometries, not only the crystal ones in Fig. 1. This would affect the accessibility of hydrophobic groups (l. 145-146) as well as the calculated chemical shifts.

    3. Regarding NMR, the experimental proton chemical-shift differences between 4’ and 4’’ are not large enough to be conclusive and the calculated ones do not fully correspond to the experiment (this is common, but I think the comparison between exp. and calc. shifts cannot be used for a definitive assignment of the stereoisomers in this case). Were some 2D correlation NMR spectra measured? Depending on the conformational exchange mentioned above, NOESY could help if the structures in Fig. 1 are the most populated ones in solution.

  2. The atom numbering is inconsistent: while in the first sections of the manuscript, 1,3,4, and 6 are the numbers of N atoms, locants 3a and 6a are used for the central pair of C atoms on l. 153. Fig. 4 and Tab. 3 and the accompanying text uses 5 and 10 for these two carbons. Fig. 1 uses completely different numbering. I would welcome a presentation of atom numbering in Scheme 1 and keeping it in the whole article. Fig. 4 would then become unnecessary.

  3. If I understood correctly, it follows from the results that a racemate of 1 was used for the synthesis but I think it should be clearly stated in the Methods or elsewhere in the text.

  4. NMR spectra of by-products are mentioned on l. 54-58. The spectrum of the reaction mixture should be shown.

  5. Scheme 3 is partially redundant with Scheme 1 and contains the following issues:

    1. hydroxy-substituents are listed as 4c and 4e, whereas these are 4e and 4f in Scheme 1;

    2. the nitro-compounds are differently from Scheme 1 as well. Please check the compound labels;

    3. the pairs in Scheme 3 are not enantiomers (having the same configuration next to P);

    4. the definition of 4’ and 4’’ stereoisomers should come earlier; could it be included in Scheme 1?

  6. The section Materials and Methods should describe the origin and purity of the chemicals used.

Minor- comments

  1. Typo in the title (phosphorous should be phosphorus)

  2. Are the locant identifiers “1” within the brackets needed on l. 10, when they refer to methyl?

  3. “Some” on l. 11 should be specified more precisely

  4. l. 24: refs. 3 and 4 are patents, can the authors provide any scientific work for reference?

  5. l. 38-40 should be better placed into Introduction and the beginning of the new results should be more clearly separated from the review of the literature.

  6. I would appreciate a review of synthetic methods used and previously published results with similar compounds in the Introduction, as well as a brief description of the chiral centers in such molecules.

  7. l. 44: “Some” is not very helpful

  8. Scheme 1: include atom numbering; “Yeild” typos; better quality of the graphics is needed (vector format); note that compound 1 is racemate (if so)

  9. l. 50: 1-aminoalkanoalkaphosphonates?

  10. l. 51: “Very” is not much precise

  11. l. 56: rather ester than ether

  12. paragraph on l. 62-65 seems redundant

  13. Table 1: Does the reaction time correspond to the maximal yield? This should be noted. The alignment in between of two rows doesn’t help me to orient in the table. Alignment to the first row and leaving the common fields in the second row empty might help.

  14. l. 93: “Thus” seems inappropriate

  15. The reaction times on l. 96-99 do not agree with rows 4-5 in Table 1

  16. l. 107: trifluoromethane acid?

  17. l. 110: should probably be “confirms”

  18. Could the authors comment more on the lower yields when using Lewis acids on l. 111-113?

  19. Table 2: epsilon should be explained

  20. It should be noted whether the conditions on l. 122-124 are the same as in Table 1

  21. l. 128-132 describe the decrease of efficiency at elevated temperatures. Have the authors tried lower temperatures?

  22. l. 147-148 use different nomenclature than the previous parts of the manuscript. It appears later in the text, too. Please make sure that this is correct and needed.

  23. l. 205 and l. 480: there are no methyl protons in CH-CH fragment

  24. The paragraphs on l. 203-218 and l. 238-267 use non-standard NMR terminology (“are registered”: appear, resonate, etc.; “towards weak/strong field”: upfield/downfield or to lower/higher shifts). This part of discussion is somewhat difficult to follow. Please consider rewriting and pointing out the main outcomes. In addition, referring to one specific carbon atom as “chiral” is not precise since there are three chiral carbon atoms in the compounds 4.

  25. l. 219-225: it should be discussed that the highest diastereomeric excess in 4f does not agree with the lowest energy difference from calculations

  26. Table 3 contains both enantiomers which is superfluous; if there are compounds for which differences are observed, it should be noted elsewhere

  27. l. 282-289: does “sensor“ mean “probehead”? Shifts are usually referenced, not “assigned” to TMS. If TMS present in the samples (as appears from supplementary figures), this should be written. Were 13C and 31P shifts calibrated directly or using calculations for the reference frequencies from 1H?

Conclusion

I think that the scientific contribution of the manuscript is worth publishing in IJMS. The compounds synthesized are novel and important in the field. The products are well characterized but the presented evidence for the assignment of the two diastereomers of 4b-f is, in my opinion, not conclusive. Furthermore, the atom numbering seems inconsistent throughout the manuscript and several errors in compound labeling were encountered. For these reasons, I recommend acceptance of the article for publishing only after major revisions.

Comments on the Quality of English Language

Although the text can be well understood, some constructions appear cumbersome (l. 12-13, l. 28-34, l. 66-69) and several words are chosen inappropriately (e.g., appliance instead of application, regularities instead of rules, aprotonic instead of aprotic). In addition, the manuscript contains relatively high number of typos and formatting issues. I recommend careful editing of the manuscript.

Author Response

Dear Reviewer,

Thank you for your attention to our manuscript. This careful review helped us to better understand the results obtained. All comments and recommendations helped us to improve the manuscript. The comments and corrections to the review are presented below.

Reviewer comment

Comments and Suggestions for Authors

The present manuscript describes the synthesis of several (diphenylphosphono)methyl derivatives of 1,3,4-trimethylglycolurile and optimal conditions for the highest possible yields of the target products are presented. The final products are separated by HPLC and characterization by XRD and NMR with respect to stereochemistry.

  1. The diastereomers 4a’ and 4a’’ were separated by HPLC and their structures were obtained from single-crystal XRD. This makes a solid ground for unambiguous interpretation of the NMR spectra. However, 4b-f were not analyzed by XRD.
    1. How can the authors prove that the 4’ vs. 4’’ stereoisomers of 4b-f are the ones that are presented in the manuscript? Is the order and separation in HPLC clear enough? The elusion profiles should be part of the publication.

Author response

Thank you for your comment. The XRD analysis of substances 4a' and 4a" shows that in a pair of diastereomers the highest retention time is characteristic of the substance in which a higher number of hydrophobic substituents can be co-directed and simultaneously interact with the surface of the stationary phase [DOI:10.1007/bf02491708]. By analogy, we assumed that for all similar diastereomers the higher retention value can be an indication of the co-directional position of the hydrophobic substituents around the asymmetric carbon atoms.

The order and separation of diastereomers in our opinion is sufficient for preparative isolation of the corresponding fractions. We also present here the fragments of chromatograms for compounds 4a-f. These figures were added to SI (Figs. 46-51).

Fig. 1 Preparative elution profile of 4a

Fig. 2 Preparative elution profile of 4b

Fig. 3 Preparative elution profile of 4c

Fig. 4 Preparative elution profile of 4d

Fig. 5 Preparative elution profile of 4e

Fig. 6 Preparative elution profile of 4f

Reviewer comment

  1. Solution conformation is not discussed. One can expect several exchanging geometries, not only the crystal ones in Fig. 1. This would affect the accessibility of hydrophobic groups (l. 145-146) as well as the calculated chemical shifts.

Author response

Thank you for your comment. We agree with the observation that different conformations can be expected in the solution. However, the experimental data are averaged signals from conformations that are in equilibrium with each other in solution.

Reviewer comment

Regarding NMR, the experimental proton chemical-shift differences between 4’ and 4’’ are not large enough to be conclusive and the calculated ones do not fully correspond to the experiment (this is common, but I think the comparison between exp. and calc. shifts cannot be used for a definitive assignment of the stereoisomers in this case). Were some 2D correlation NMR spectra measured? Depending on the conformational exchange mentioned above, NOESY could help if the structures in Fig. 1 are the most populated ones in solution.

Author response

Thank you for your comment. We performed an additional NOESY experiment for compounds 4a' and 4a''. Indeed, interactions between protons 3a,6a and CHPO3 are observed for compound 4a' (Fig 7). At the same time, these interactions were not detected for compound 4a'' (Fig 8).

Fig 7. 2D NOESY spectrum of 4a’

Fig 8. 2D NOESY spectrum of 4a”

Reviewer comment

  1. The atom numbering is inconsistent: while in the first sections of the manuscript, 1,3,4, and 6 are the numbers of N atoms, locants 3a and 6a are used for the central pair of C atoms on l. 153. Fig. 4 and Tab. 3 and the accompanying text uses 5 and 10 for these two carbons. Fig. 1 uses completely different numbering. I would welcome a presentation of atom numbering in Scheme 1 and keeping it in the whole article. Fig. 4 would then become unnecessary.

Author response

Thank you for your comment. We have modified scheme 1 by adding the numbering of the atoms under discussion and deleted figure 4.

Reviewer comment

  1. If I understood correctly, it follows from the results that a racemate of 1 was used for the synthesis but I think it should be clearly stated in the Methods or elsewhere in the text.

Author response

Thank you for your comment. Yes, compound 1 was used as a racemate. We have also added a clarification in Scheme 1 and in the Materials and methods section.

Reviewer comment

  1. NMR spectra of by-products are mentioned on l. 54-58. The spectrum of the reaction mixture should be shown.

Author response

Thank you for your comment. The typo of compound 7a was corrected to 6a. We added 31P spectrum of the reaction mixture to SI (Fig. 1).

Reviewer comment

Scheme 3 is partially redundant with Scheme 1 and contains the following issues:

  1. hydroxy-substituents are listed as 4c and 4e, whereas these are 4e and 4f in Scheme 1;

Author response

Thank you for your comment. Scheme 1 was corrected.

Reviewer comment

  1. the nitro-compounds are differently from Scheme 1 as well. Please check the compound labels;

Author response

Thank you for your comment. Scheme 1 was revised.

Reviewer comment

  1. the pairs in Scheme 3 are not enantiomers (having the same configuration next to P);

Author response

Thank you for your comment. Indeed, the pairs in Scheme 3 are not enantiomers. Scheme 1 was revised. Scheme 3 was removed.

Reviewer comment

  1. the definition of 4’ and 4’’ stereoisomers should come earlier; could it be included in Scheme

Author response

Thank you for your comment. We have introduced the designations 4' and 4'' in Scheme 1.

Reviewer comment

  1. The section Materials and Methods should describe the origin and purity of the chemicals used.

Author response

Thank you for your comment. We added information on solvents and reagents used to section 4 "Materials and Methods”.

Reviewer comment

Minor- comments

  1. Typo in the title (phosphorous should be phosphorus)

Author response

Thank you for your comment. The typo was corrected.

Reviewer comment

  1. Are the locant identifiers “1” within the brackets needed on l. 10, when they refer to methyl?

Author response

Thank you for your comment. Redundant locant identifiers were removed from the common name of the compounds.

Reviewer comment

  1. “Some” on l. 11 should be specified more precisely

Author response

Thank you for your comment. In line 11, the word "some" was removed, and the specific 2a-f benzaldehydes were listed.

Reviewer comment

  1. 24: refs. 3 and 4 are patents, can the authors provide any scientific work for reference?

Author response

Thank you for the comment. According to our data, the results described in these patents were not presented in scientific articles.

Reviewer comment

  1. 38-40 should be better placed into Introduction and the beginning of the new results should be more clearly separated from the review of the literature.

Author response

Thank you for the comment. The lines 38-40 were moved to the introduction section.

Reviewer comment

  1. I would appreciate a review of synthetic methods used and previously published results with similar compounds in the Introduction, as well as a brief description of the chiral centers in such molecules.

Author response

Thank you for the comment. A description of phosphorus-containing glycoluryl derivatives with references (DOI:10.1134/s1070427216010213, ref [18]) and DOI:10.1002/chem.201902556 [19]) was added to the introduction section.

Reviewer comment

  1. 44: “Some” is not very helpful

Author response

Thank you for the comment. "Some" was removed from the sentence.

Reviewer comment

  1. Scheme 1: include atom numbering; “Yeild” typos; better quality of the graphics is needed (vector format); note that compound 1 is racemate (if so)

Author response

Thank you for the comment. Yes, we used compound 1 as a racemate. Information about this was added in Scheme 1 and also in Materials and Methods section.

Reviewer comment

  1. 50: 1-aminoalkanoalkaphosphonates?

Author response

Thank you for the comment. The typo was corrected.

Reviewer comment

  1. 51: “Very” is not much precise

Author response

Thank you for the comment. The sentence was revised.

Reviewer comment

  1. 56: rather ester than ether

Author response

Thank you for the comment. The sentence was revised.

Reviewer comment

  1. paragraph on l. 62-65 seems redundant

Author response

Thank you for the comment. This paragraph was deleted from the text of the manuscript.

Reviewer comment

  1. Table 1: Does the reaction time correspond to the maximal yield? This should be noted. The alignment in between of two rows doesn’t help me to orient in the table. Alignment to the first row and leaving the common fields in the second row empty might help.

Author response

Thank you for the comment. A clarification on the maximal yield was added to the Description of Table 1. Table 1 was formatted accordingly.

Reviewer comment

  1. 93: “Thus” seems inappropriate

Author response

Thank you for the comment. The sentence was revised.

Reviewer comment

  1. The reaction times on l. 96-99 do not agree with rows 4-5 in Table 1

Author response

Thank you for the comment. The values in row 4 in Table 1 were corrected. An inaccuracy in lines 96-98 was corrected. We meant methanesulfonic acid, not trifluoromethanesulfonic acid.

Reviewer comment

  1. 107: trifluoromethane acid?

Author response

Thank you for the comment. We replaced "trifluoromethane acid" by "trifluoromethanesulfonic acid".

Reviewer comment

  1. 110: should probably be “confirms”

Author response

Thank you for the comment. The sentence was revised.

Reviewer comment

  1. Could the authors comment more on the lower yields when using Lewis acids on l. 111-113?

Author response

Thank you for the comment. According to Ref. [10.1021/jo00916a019], the reaction proceeds via two stages. In the first stage, the amide interacts with the aldehyde to form the corresponding iminium, which reacts with the triphenylphosphite in the second stage. Both Brønsted and Lewis acids exhibit catalytic activity in this reaction, however, as shown in Ref. [10.1080/10426500008045250], the Brønsted acids are preferred for the first stage to proceed, while the Lewis acids catalyze the second stage of the phosphonate formation. In view of the fact that 1,3,4-trimethylglycoluryl is less reactive than the unsubstituted carbamates, the percolation of the iminium formation step is limiting. The use of strong Brønsted acids is preferred in this case, which is consistent with the yields given in Table 1.

Reviewer comment

  1. Table 2: epsilon should be explained

Author response

Thank you for the comment. An explanation was added into Table 2.

Reviewer comment

  1. It should be noted whether the conditions on l. 122-124 are the same as in Table 1

Author response

Thank you for the comment. A clarification was added to the text of the article in lines 114-115 in revised manuscript.

Reviewer comment

  1. 128-132 describe the decrease of efficiency at elevated temperatures. Have the authors tried lower temperatures?

Author response

Thank you for the comment. No, we did not lower the temperature below room temperature, as in our opinion a yield of 82 % is acceptable.

Reviewer comment

  1. 147-148 use different nomenclature than the previous parts of the manuscript. It appears later in the text, too. Please make sure that this is correct and needed.

Author response

Thank you for the comment. We changed "diphenyl (phenyl(3,4,6-trimethyl-2,5-dioxohexahydroimidazo[4,5-d]imidazol-1(2H)-yl)methyl)phosphonate" to "4a′ and 4a″" in lines 146 (revised manuscript). We also left the common name of the obtained compounds 1-[aryl-(diphenylphosphono)methyl]-3,4,6-trimethylglycolurils in both Abstract and Conclusion sections. In our opinion, this common name is easier to understand than the IUPAC name.

Reviewer comment

  1. 205 and l. 480: there are no methyl protons in CH-CH fragment

Author response

Thank you for the comment. The sentence was revised.

Reviewer comment

  1. The paragraphs on l. 203-218 and l. 238-267 use non-standard NMR terminology (“are registered”: appear, resonate, etc.; “towards weak/strong field”: upfield/downfield or to lower/higher shifts). This part of discussion is somewhat difficult to follow. Please consider rewriting and pointing out the main outcomes. In addition, referring to one specific carbon atom as “chiral” is not precise since there are three chiral carbon atoms in the compounds 4.

Author response

Thank you for the comment. The terms used were revised throughout the manuscript.

Reviewer comment

  1. 219-225: it should be discussed that the highest diastereomeric excess in 4f does not agree with the lowest energy difference from calculations

Author response

Thank you for the comment. Indeed, the experimental data indicate an excess of isomer 4f, while the calculated changes in electronic energy and enthalpy for the process 4f → 4f are the smallest in comparison with the other derivatives. This fact may be related to the accuracy of DFT energy characterization (~1 kcal/mol, https://pubs.rsc.org/en/content/articlelanding/2019/sc/c9sc02834j), and also the ratio may be determined by the energy barriers of the transition of one diastereomer to another, the labour- and resource-intensive search for which was not carried out.

Reviewer comment

  1. Table 3 contains both enantiomers which is superfluous; if there are compounds for which differences are observed, it should be noted elsewhere

Author response

Thank you for the comment. Table 3 has been modified according to your recommendations.

Reviewer comment

  1. 282-289: does “sensor“ mean “probehead”? Shifts are usually referenced, not “assigned” to TMS. If TMS present in the samples (as appears from supplementary figures), this should be written. Were 13C and 31P shifts calibrated directly or using calculations for the reference frequencies from 1H?

Author response

Thank you for the comment. We have corrected the terminology in lines 282-289. δ 13C in the region of 0 ppm are signals related to acetonitrile. Since the substances mostly represent viscous oil-like samples, we were not able to completely get rid of the acetonitrile impurity. We also note that the compound 4a' also contains an acetonitrile molecule as a solvate. The reference frequencies from 1H were using to calibrated 13C and 31P shifts.

Reviewer comment

Conclusion

I think that the scientific contribution of the manuscript is worth publishing in IJMS. The compounds synthesized are novel and important in the field. The products are well characterized but the presented evidence for the assignment of the two diastereomers of 4b-f is, in my opinion, not conclusive. Furthermore, the atom numbering seems inconsistent throughout the manuscript and several errors in compound labeling were encountered. For these reasons, I recommend acceptance of the article for publishing only after major revisions.

Author response

Thank you for your attention to our manuscript.

Reviewer comment

Comments on the Quality of English Language

 Although the text can be well understood, some constructions appear cumbersome (l. 12-13, l. 28-34, l. 66-69) and several words are chosen inappropriately (e.g., appliance instead of application, regularities instead of rules, aprotonic instead of aprotic). In addition, the manuscript contains relatively high number of typos and formatting issues. I recommend careful editing of the manuscript.

Author response

Thank you for your comment. English was proof-read by a native speaker.

Sincerely yours and on behalf of the authors,

Vera P. Tuguldurova

Reviewer 3 Report

Comments and Suggestions for Authors

The paper refers to the synthesis of novel phosphorous-containing derivatives of 1,3,4-2 trimethylglycolurile by Birum-Oleksyszyn reaction. The isolated diastereomers were characterized by 1H, 13C and 31P NMR, and the structures of some diastereomers were confirmed by single-crystal X-ray crystal structure analysis. The authors investigated the influence of synthesis conditions on the yield of phosphonate derivatives of 1,3,4-trimethylglycoluriles and the separation and identification of diastereomers of phosphonate derivatives of 1,3,4-trimethylglycoluriles. The paper is generally well organized, however, its content is not related to the objectives of the Special Issue ‘Application of NMR Spectroscopy in Biomolecules’. The article could be published in Molecules after major revision as indicated below:

1. Page 5, lines 152 & 153.

 “…metric, therefore, both enantiomeric forms of compound 4a′ crystallize as a true racemate. 152 The configuration of the stereocenters (3a, 6a, CHPO3) in compound 4a′ is as follows: R,S,R….”.

The notations of the stereocenters are different than those denoted in Table 3. A uniform notation should be used throughout the text, Tables and Figures.

2. Page 9, lines 252-255.

 Comparison of experimental and calculated δ values of hydrogen atoms at the chiral carbon atom of the phosphonate substituent (12C) as well as the bridging CH-CH (5C, 10C) group indicates their significant divergence for enantiomeric pairs of RSR and SRS (4″) type and the absence of such divergence in RRS and SSR (4′) pairs”.

Comparison of the experimental calculated δ values should not be based only on protons of one to three chiral carbons but considering all the chemical shifts. The conclusions should be based on the statistical analysis of the resulting correlations.

3. Page 9, 262-267.

 The above effect is not observed in the experimental 1H NMR spectra, which may be due to the inability to account for the solvating influence of the solvent by ab initio methods and the corresponding change in the conformations of the structures in real solution. The observed influence of phenyl rings in RSR- and SRS-type enantiomers is retained for OH- and NO2- derivatives (4b″-4f″)e”.

The authors should specify whether they investigated the low energy conformers after a rotation of the single bonds at least of two of the substituents. See also my comment below.

4.3.6. Computational details, lines 448-450.

 Geometry optimization of all the structures was carried out using the Gaussian’09 program package [25] installed on the SKIF “Cyberia” supercomputer of Tomsk State University. Hybrid functional M062X [26] and split-valence basis set 6-311+G(2d,p)”.

As noted above, it should be specified whether geometry optimizations involved the X-ray structure determination without the complete conformational analysis around single bonds.

5. Page 15, lines 484 & 485.

signal at the chiral atom of the phosphonate group is shifted to a stronger field relative to the analogous proton of the 4″ structure”.

The term “stronger field” should be avoided and the notation “lower or higher δ values” or “more shielding/less shielding” should be utilized.

6. Figure S14.

The lineshape of OH group at ~9.6 ppm is asymmetric. Very probably, the solution was not homogeneous enough. The spectrum could be recorded again.

Author Response

Dear Reviewer,

Thank you for your attention to our manuscript. This careful review helped us to better understand the results obtained. All comments and recommendations helped us to improve the manuscript. The comments and corrections to the review are presented below.

Reviewer comment

Comments and Suggestions for Authors

The paper refers to the synthesis of novel phosphorous-containing derivatives of 1,3,4-2 trimethylglycolurile by Birum-Oleksyszyn reaction. The isolated diastereomers were characterized by 1H, 13C and 31P NMR, and the structures of some diastereomers were confirmed by single-crystal X-ray crystal structure analysis. The authors investigated the influence of synthesis conditions on the yield of phosphonate derivatives of 1,3,4-trimethylglycoluriles and the separation and identification of diastereomers of phosphonate derivatives of 1,3,4-trimethylglycoluriles. The paper is generally well organized, however, its content is not related to the objectives of the Special Issue ‘Application of NMR Spectroscopy in Biomolecules’. The article could be published in Molecules after major revision as indicated below:

  1. Page 5, lines 152 & 153.

 “…metric, therefore, both enantiomeric forms of compound 4a′ crystallize as a true racemate. 152 The configuration of the stereocenters (3a, 6a, CHPO3) in compound 4a′ is as follows: R,S,R….”.

The notations of the stereocenters are different than those denoted in Table 3. A uniform notation should be used throughout the text, Tables and Figures.

Author response

Thank you for your comment. The notations were revised accordingly.

Reviewer comment

  1. Page 9, lines 252-255.

 “Comparison of experimental and calculated δ values of hydrogen atoms at the chiral carbon atom of the phosphonate substituent (12C) as well as the bridging CH-CH (5C, 10C) group indicates their significant divergence for enantiomeric pairs of RSR and SRS (4″) type and the absence of such divergence in RRS and SSR (4′) pairs”.

Comparison of the experimental calculated δ values should not be based only on protons of one to three chiral carbons but considering all the chemical shifts. The conclusions should be based on the statistical analysis of the resulting correlations.

Author response

Thank you for your comment. The obtained dependences of the chemical shift values of stereocentre protons on the type of diastereomer are qualitative, thus, it is rather difficult to perform statistical analysis. We added a comparison of the chemical shift values of chiral carbon atoms and phosphorus atoms into Table 3. The correlation plots of the experimental and calculated values of these signals are also given in the Supplementary Information.

Reviewer comment

  1. Page 9, 262-267.

 “The above effect is not observed in the experimental 1H NMR spectra, which may be due to the inability to account for the solvating influence of the solvent by ab initio methods and the corresponding change in the conformations of the structures in real solution. The observed influence of phenyl rings in RSR- and SRS-type enantiomers is retained for OH- and NO2- derivatives (4b″-4f″)e”.

The authors should specify whether they investigated the low energy conformers after a rotation of the single bonds at least of two of the substituents. See also my comment below.

  1. “3.6. Computational details, lines 448-450.

 “Geometry optimization of all the structures was carried out using the Gaussian’09 program package [25] installed on the SKIF “Cyberia” supercomputer of Tomsk State University. Hybrid functional M062X [26] and split-valence basis set 6-311+G(2d,p)”.

As noted above, it should be specified whether geometry optimizations involved the X-ray structure determination without the complete conformational analysis around single bonds.

Author response

Thank you for your comment. In this work, a full conformational analysis of all structures was not performed. Geometry optimization was performed based on the experimentally determined X-ray structures 4a' and 4a'' without performing conformational analysis, which is a common practice (see DOI:10.1016/j.molstruc.2022.134669; DOI:10.1016/j.jics.2022.100437). A clarification regarding the lack of conformational analysis was added the Computational details section of the manuscript.

Reviewer comment

  1. Page 15, lines 484 & 485.

signal at the chiral atom of the phosphonate group is shifted to a stronger field relative to the analogous proton of the 4″ structure”.

The term “stronger field” should be avoided and the notation “lower or higher δ values” or “more shielding/less shielding” should be utilized.

Author response

Thank you for your comment. The terms were revised accordingly.

Reviewer comment

  1. Figure S14.

The lineshape of OH group at ~9.6 ppm is asymmetric. Very probably, the solution was not homogeneous enough. The spectrum could be recorded again.

Author response

Thank you for your comment. We re-recorded the 1Н spectrum of this compound. The form of signal is due to the presence of compound 4c″ as a minor impurity, the OH-group of which has a close chemical shift. Figure 1 shows the superimposed 1Н NMR spectra of compound 4c′ (red) and compound 4c″ (blue).

Figure 1. Superimposed 1H NMR spectra of compound 4c′ (red) and compound 4c″ (blue)

Sincerely yours and on behalf of the authors,

Vera P. Tuguldurova

Reviewer 4 Report

Comments and Suggestions for Authors

ijms-2636069

Reviewer Report

In this paper, the synthesis of novel phosphorus-containing derivatives of 1,3,4-trimethylglycolurile using the Birum-Oleksyszyn reaction is reported. Results of the study on the influence of the reaction conditions (various substituted benzaldehydes, solvents, catalysts, temperature, reaction time) are given. The isolated diastereomers were characterized by NMR spectroscopy, and the crystal structure of the diastereomers of the reaction with benzaldehyde, determined by single crystal X-ray analysis, is given. To distinguish the diastereomers, three 1H-NMR signals were selected and their shift values were compared with the quantum chemically calculated shielding constants (chemical shifts).

Since the paper is to be published in the special issue of the journal “Application of NMR Spectroscopy in Biomolecules”, I wonder why only the 1H chemical shifts of three selected H-atoms were assigned and discussed, namely the H-atom bonded to the chiral C-atom and the two H-atoms of the CH-CH bridge. The NMR spectra are only partially evaluated. The paper would be much more valuable if the given NMR data were assigned to the corresponding atoms.

Looking at the 31P and 13C chemical shifts values, it becomes clear that the signal of the P- atom and that of the chiral C-atom are best suited to distinguish the diastereomers. Here the chemical shift differences show a characteristic dependence. Why don’t you consider the chemical shift of the chiral C-atom and the P-atom attached to it in your discussion. Here it is easy to distinguish between the two diastereomers 4’ and 4’’. Add a table with these results and discuss them. For examining a reaction mixture, 31P NMR spectroscopy is much more suitable than the 1H NMR spectroscopy.

What is the goal/purpose of quantum chemical calculations? As shown and described, the NMR spectroscopy enables reliable identification of the diastereomers.

The conclusion that the quantum-chemical calculations agree with the experimental data is difficult to understand. In particular, for the compounds 4’’, large differences are observed not only for δ(1H) of the 5C proton, but also for the 10C and 12C protons.

Some typographical errors make the work difficult to read. A thorough correction is required.

Line 45: In Scheme 1 it must be called yield.

Lines 54-56: 31P chemical shift values between 4.48 and 1.43 ppm are more typical for phosphoric esters or diphosphonic acids, but not for monophosphonic acids. You describe a partial or complete hydrolysis of ether groups. Do you mean ester groups?

Line 90: Replace appliance with application.

Lines 153-154: A structural formula in which the three stereocenters are marked would be helpful at this point.

Line 205: There are no methyl protons in the CH-CH bond. The single proton is part of a methanetriyl group (the term methine group is also used).

Line 220: Insert a reference to the supplementary material here.

Line 228: It must read diastereomers.

Line 460: Tables show …

Author Response

Dear Reviewer,

Thank you for your attention to our manuscript. This careful review helped us to better understand the results obtained. All comments and recommendations helped us to improve the manuscript. The comments and corrections to the review are presented below.

Reviewer comment

In this paper, the synthesis of novel phosphorus-containing derivatives of 1,3,4-trimethylglycolurile using the Birum-Oleksyszyn reaction is reported. Results of the study on the influence of the reaction conditions (various substituted benzaldehydes, solvents, catalysts, temperature, reaction time) are given. The isolated diastereomers were characterized by NMR spectroscopy, and the crystal structure of the diastereomers of the reaction with benzaldehyde, determined by single crystal X-ray analysis, is given. To distinguish the diastereomers, three 1H-NMR signals were selected and their shift values were compared with the quantum chemically calculated shielding constants (chemical shifts).

Since the paper is to be published in the special issue of the journal “Application of NMR Spectroscopy in Biomolecules”, I wonder why only the 1H chemical shifts of three selected H-atoms were assigned and discussed, namely the H-atom bonded to the chiral C-atom and the two H-atoms of the CH-CH bridge. The NMR spectra are only partially evaluated. The paper would be much more valuable if the given NMR data were assigned to the corresponding atoms.

Looking at the 31P and 13C chemical shifts values, it becomes clear that the signal of the P- atom and that of the chiral C-atom are best suited to distinguish the diastereomers. Here the chemical shift differences show a characteristic dependence. Why don’t you consider the chemical shift of the chiral C-atom and the P-atom attached to it in your discussion. Here it is easy to distinguish between the two diastereomers 4’ and 4’’. Add a table with these results and discuss them. For examining a reaction mixture, 31P NMR spectroscopy is much more suitable than the 1H NMR spectroscopy.

Author response

Thank you for your comment. We added a comparison of the chemical shift values of chiral carbon atoms and phosphorus atoms into Table 3. The correlation plots of the experimental and calculated values of these signals are also given in the Supplementary Information.

Reviewer comment

What is the goal/purpose of quantum chemical calculations? As shown and described, the NMR spectroscopy enables reliable identification of the diastereomers.

Author response

Thank you for your comment. The obtained results of quantum-chemical calculations allow us to unambiguously identify the structure of diastereomers without performing time-consuming analyses (PCA), and additionally confirm the energetic stability of the structures and can predict the diastereomeric ratio of new phosphorus-containing derivatives of glycoluril.

Reviewer comment

The conclusion that the quantum-chemical calculations agree with the experimental data is difficult to understand. In particular, for the compounds 4’’, large differences are observed not only for δ(1H) of the 5C proton, but also for the 10C and 12C protons.

Author response

Thank you for your comment. The agreement of quantum-chemical calculations with experimental data in this work consists not only in comparison of the values of chemical shifts of atoms in the structures, but also in confirmation of greater or lesser stability of diastereomers on the basis of calculated energy and thermodynamic characteristics of the structures. It is shown that the diastereomers of the 4'' type are more stable, which is confirmed by the experimental relations with the excess of 4'' structures in the products.

Reviewer comment

Some typographical errors make the work difficult to read. A thorough correction is required.

Author response

Thank you for your comment. The errors were corrected throughout the manuscript.

Reviewer comment

Line 45: In Scheme 1 it must be called yield.

Author response

Thank you for your comment. The typo was corrected.

Reviewer comment

Lines 54-56: 31P chemical shift values between 4.48 and 1.43 ppm are more typical for phosphoric esters or diphosphonic acids, but not for monophosphonic acids. You describe a partial or complete hydrolysis of ether groups. Do you mean ester groups?

Author response

Thank you for your comment. Yes, we meant ester groups. The sentence was revised (Lines 55-58) and added the 31P spectrum of the reaction mixture to the supplementary materials (Fig 1)

Reviewer comment

Line 90: Replace appliance with application.

Author response

Thank you for your comment. The sentence was revised.

Reviewer comment

Lines 153-154: A structural formula in which the three stereocenters are marked would be helpful at this point.

Author response

Thank you for your comment. We added the designations of stereocentres to Scheme 1 for convenience.

Reviewer comment

Line 205: There are no methyl protons in the CH-CH bond. The single proton is part of a methanetriyl group (the term methine group is also used).

Author response

Thank you for your comment. The typo was corrected.

Reviewer comment

Line 220: Insert a reference to the supplementary material here.

Author response

Thank you for your comment. The comment is not clear to us. Is it necessary to refer to the figure S45 indicating the presence of a mass spectrum only for compound 4f"?

Reviewer comment

Line 228: It must read diastereomers.

Author response

Thank you for your comment. The typo was corrected.

Reviewer comment

Line 460: Tables show …

Author response

Thank you for your comment. The sentence was revised.

Sincerely yours and on behalf of the authors,

Vera P. Tuguldurova

Round 2

Reviewer 2 Report

Comments and Suggestions for Authors

Dear Author's,

thank you for addressing carefully all the issues raised during the first review round. Especially, I appreciate the experimental confirmation of diastereomers 4a'/4a'' by NOESY and I would suggest to include these 2D spectra in SI.

In my opinion, the readability and the overall look of the manuscript have considerably increased. However, the quality of Fig. 1 should be improved by removing the JPEG-looking compression and resolution enhancement. In addition, the asterisks that mark stereocenters should be explained in the caption of Fig. 1.

I recommend the article for publication after these minor corrections are made.

Author Response

Dear Reviewer,

Thank you for your attention to our manuscript. The comments and corrections to the review are presented below.

Reviewer comment

Dear Author's,

thank you for addressing carefully all the issues raised during the first review round. Especially, I appreciate the experimental confirmation of diastereomers 4a'/4a'' by NOESY and I would suggest to include these 2D spectra in SI.

Author response

Thank you for your comment. We added information about the NOESY experiment to the text of the paper (lines 268-271). The NOESY spectra of compounds 4a'/4a'' were added to SI (Fig S35-S36).

Reviewer comment

In my opinion, the readability and the overall look of the manuscript have considerably increased. However, the quality of Fig. 1 should be improved by removing the JPEG-looking compression and resolution enhancement. In addition, the asterisks that mark stereocenters should be explained in the caption of Fig. 1.

I recommend the article for publication after these minor corrections are made.

Author response

We also improved the quality of Fig. 1 and added a clarification of asterisks in the caption of Fig. 1.

Thank you for your opinion about our work and for your comments.

Sincerely yours and on behalf of the authors,

Vera P. Tuguldurova

Reviewer 3 Report

Comments and Suggestions for Authors

The authors have successfully incorporated the majority of the requested additions and corrections. One of issue of concern, however, is that the authors did not provide the complete conformational analysis around single bonds at least for one case. Therefore, further revision is requested.

Author Response

Dear Reviewer,

Thank you for your attention to our manuscript. The comments and corrections to the review are presented below.

Reviewer comment

The authors have successfully incorporated the majority of the requested additions and corrections. One of issue of concern, however, is that the authors did not provide the complete conformational analysis around single bonds at least for one case. Therefore, further revision is requested.

Author response

Thank you for your comment. We performed conformational analysis using the relaxed scan around the single bond C-P (rotation of the phosphonate fragment) and around the single bond C-N (rotation of the glycoluryl fragment) in 15° steps for all structures of compound 4a at the R2SCAN-3C level of theory. The results obtained are presented in tables 1 and 2 and figures 1-8.

Table 1. Results of conformational analysis around the single bond (C-P) of the 4a

Structure

The dihedral angle relative to the initial position, °

The dihedral angle (NCPO), °

∆E, kcal/mol

δ 1H of CHPO3 proton, ppm

δ 1H of 6a proton, ppm

δ 1H of 3a proton, ppm

δ 13C of CHPO3, ppm

δ 13C of 6a, ppm

δ 13C of 3a, ppm

δ 31P, ppm

4a′ (RSR)

Experimental chemical shifts

5.77

5.28

5.24

57.37

72.4

72.14

14.06

0

41

0.00

5.64

5.47

5.41

59.69

72.80

71.71

19.31

15

56

1.32

30

71

2.73

45

86

3.33

60

101

5.65

75

116

6.90

90

131

2.98

105

146

1.42

120

161

1.47

135

176

1.01

6.88

4.68

5.21

54.13

66.78

73.21

19.84

150

191

1.65

165

206

2.53

180

221

3.37

195

236

3.87

210

251

3.47

225

266

3.34

240

281

2.70

255

296

2.19

270

311

1.64

6.93

4.73

5.21

58.16

67.58

73.47

22.38

285

326

2.92

300

341

4.45

315

356

4.33

330

371

4.24

345

386

0.48

7.03

5.72

5.47

57.21

66.53

72.65

26.75

360

401

0.60

4a′ (SRS)

Experimental chemical shifts

5.77

5.28

5.24

57.37

72.4

72.14

14.06

0

-41

0.49

5.64

5.47

5.41

59.69

72.80

71.71

19.31

15

-26

0.86

30

-11

2.45

45

4

4.06

60

19

5.21

75

34

5.00

90

49

3.48

105

64

2.95

120

79

3.02

135

94

2.00

5.81

5.47

5.52

61.03

73.47

72.12

19.63

150

109

2.20

165

124

4.00

180

139

5.62

195

154

5.95

210

169

4.12

225

184

3.47

240

199

2.88

255

214

0.00

5.72

5.40

5.54

58.22

72.29

72.33

20.20

270

229

1.44

285

244

2.81

300

259

4.57

315

274

4.77

330

289

3.44

345

304

1.85

360

319

0.49

5.61

5.46

5.43

60.18

73.13

72.11

19.27

4a″ (SRR)

Experimental chemical shifts

5.85

5.50

5.00

53.84

70.38

72.32

13.35

0

57

0.00

6.88

5.32

3.18

51.98

70.20

70.26

17.52

15

72

0.98

30

87

1.22

45

102

2.75

60

117

3.55

75

132

4.03

90

147

3.45

105

162

3.07

120

177

2.68

135

192

2.66

6.09

6.06

5.64

55.21

68.69

73.54

23.28

150

207

3.00

165

222

3.41

180

237

3.46

195

252

3.50

210

267

3.19

225

282

2.79

240

297

1.92

255

312

1.41

6.33

6.00

5.80

54.93

68.19

72.39

24.59

270

327

2.02

285

342

2.53

300

357

2.78

315

372

2.93

330

387

2.96

345

402

2.44

360

417

1.67

6.70

6.14

5.55

54.05

69.04

72.65

19.35

4a″ (RSS)

Experimental chemical shifts

5.85

5.50

5.00

53.84

70.38

72.32

13.35

0

-57

0

6.88

5.32

3.18

51.98

70.20

70.26

17.52

15

-42

1.46

30

-27

2.72

45

-12

3.06

60

3

4.95

75

18

2.52

90

33

1.99

105

48

1.43

6.32

5.98

5.81

54.93

68.33

72.53

24.39

120

63

1.90

135

78

2.86

150

93

3.21

165

108

3.53

180

123

3.56

195

138

3.42

210

153

2.97

225

168

2.67

240

183

2.65

5.89

6.20

5.65

54.31

68.59

72.79

25.83

255

198

2.98

270

213

3.17

285

228

3.49

300

243

3.90

315

258

2.72

330

273

1.24

345

288

0.40

6.56

5.96

4.85

54.68

67.88

71.50

21.12

360

303

0.96

The dihedral angle relative to the initial position 0°

The dihedral angle relative to the initial position 135°

The dihedral angle relative to the initial position 270°

The dihedral angle relative to the initial position 345°

Figure 1. Results of conformational analysis around the single bond (C-P) of the structure 4a′ (RSR)

The dihedral angle relative to the initial position 0°

The dihedral angle relative to the initial position 135°

The dihedral angle relative to the initial position 255°

The dihedral angle relative to the initial position 360°

Figure 2. Results of conformational analysis around the single bond (C-P) of the structure 4a′ (SRS)

The dihedral angle relative to the initial position 0°

The dihedral angle relative to the initial position 135°

The dihedral angle relative to the initial position 255°

The dihedral angle relative to the initial position 360°

Figure 3. Results of conformational analysis around the single bond (C-P) of the structure 4a″ (SRR)

The dihedral angle relative to the initial position 0°

The dihedral angle relative to the initial position 105°

The dihedral angle relative to the initial position 240°

The dihedral angle relative to the initial position 345°

Figure 4. Results of conformational analysis around the single bond (C-P) of the structure 4a″ (RSS)

Table 1 and Figures 1-4 show that the most stable (low energy) conformations remain structures based on the experimental data (XRD) with the exception of conformation 4a′ (SRS) with a dihedral angle relative to the initial position of 255°. Optimization of this structure performed at the M062X/6-311+G(2d,p) level of theory shows that its electronic energy is lower than the electronic energy of the initial structure by 1.3 kcal/mol, which is associated with the occurrence of π-stacking between phenyl rings of the phosphonate substituent.

The main aim of the quantum chemical calculations performed in this work is to further confirm the possibility of identifying diastereomers of compounds based on NMR data. The enantiomer pairs (RSR/SRS) or (SRR/RSS) for diastereomers 4a′ and 4a″ have the same values for all chemical shifts in Table 3 of the manuscript, since these structures are mirror images of each other. The choice of other stable conformations within each enantiomer leads to different values of δ and in this case the chosen structure is not enantiomer with its pair. Thus, after the analysis, the most stable conformers remain the already described structures and the obtained δ values for such structures are sufficient to identify diastereomers, since they are in the correlation with the experimental data.

Table 2. The results of conformational analysis around the single bond (N-C) of the 4a

Structure

The dihedral angle relative to the initial position, °

The dihedral angle (CNCP), °

∆E, kcal/mol

4a′ (RSR)

0

40

0.00

15

55

0.27

30

70

1.92

45

85

3.06

60

100

4.08

75

115

4.79

90

130

5.29

105

145

4.27

120

160

4.31

135

175

2.65

150

190

1.72

165

205

1.71

180

220

1.79

195

235

1.68

210

250

1.17

225

265

1.33

240

280

0.41

255

295

2.20

270

310

7.10

285

325

13.03

300

340

6.30

315

355

5.19

330

370

3.85

345

385

1.92

360

400

0.86

4a′ (SRS)

0

-40

0.51

15

-25

1.12

30

-10

1.96

45

5

2.89

60

20

2.71

75

35

3.79

90

50

5.71

105

65

6.89

120

80

1.94

135

95

1.89

150

110

1.66

165

125

2.65

180

140

4.33

195

155

4.64

210

170

4.33

225

185

5.66

240

200

6.79

255

215

6.30

270

230

5.91

285

245

5.10

300

260

4.11

315

275

3.94

330

290

1.27

345

305

0.00

360

320

1.56

4a″ (SRR)

0

-112

0.00

15

-97

0.51

30

-82

1.05

45

-67

2.42

60

-52

4.13

75

-37

3.07

90

-22

3.04

105

-7

3.06

120

8

2.88

135

23

2.17

150

38

0.86

165

53

0.64

180

68

0.91

195

83

2.49

210

98

2.04

225

113

4.42

240

128

8.53

255

143

13.02

270

158

12.76

285

173

10.05

300

188

7.98

315

203

7.26

330

218

2.03

345

233

2.11

360

248

1.48

4a″ (RSS)

0

112

0.00

15

127

0.51

30

142

2.25

45

157

3.90

60

172

6.09

75

187

10.46

90

202

8.74

105

217

8.13

120

232

7.94

135

247

6.45

150

262

4.91

165

277

3.41

180

292

2.66

195

307

1.80

210

322

1.45

225

337

2.23

240

352

2.43

255

367

2.46

270

382

2.62

285

397

3.08

300

412

4.04

315

427

5.38

330

442

2.51

345

457

2.81

360

472

2.94

The dihedral angle relative to the initial position 0°

The dihedral angle relative to the initial position 240°

The dihedral angle relative to the initial position 360°

Figure 5. Results of conformational analysis around the single bond (C-N) of the structure 4a′ (RSR)

The dihedral angle relative to the initial position 0°

The dihedral angle relative to the initial position 150°

The dihedral angle relative to the initial position 345°

Figure 6. Results of conformational analysis around the single bond (C-N) of the structure 4a′ (SRS)

The dihedral angle relative to the initial position 0°

The dihedral angle relative to the initial position 165°

The dihedral angle relative to the initial position 360°

Figure 7. Results of conformational analysis around the single bond (C-N) of the structure 4a″ (SRR)

The dihedral angle relative to the initial position 0°

The dihedral angle relative to the initial position 210°

The dihedral angle relative to the initial position 330°

Figure 8. Results of conformational analysis around the single bond (C-N) of the structure 4a″ (RSS)

Table 2 and Figures 5-8 show that the most stable (low energy) conformations remain structures based on the experimental data (XRD) with the exception of conformation 4a′ (SRS) with a dihedral angle relative to the initial position of 345 °, which is lower by 0.5 kcal/mol, which is due to the occurrence of π-stacking between phenyl rings of the phosphonate substituent

The presented results showed that in this case a detailed conformational analysis is redundant. It is sufficient to determine the values of chemical shifts of one stable conformation to establish the type of diastereomer.

Thank you for your opinion about our work and for your comments.

Sincerely yours and on behalf of the authors,

Vera P. Tuguldurova

Reviewer 4 Report

Comments and Suggestions for Authors

The manuscript has been improved. Most of the reviewer's comments were considered. 

Author Response

Dear Reviewer,

Thank you for your attention to our manuscript. The comments and corrections to the review are presented below.

Reviewer comment

The manuscript has been improved. Most of the reviewer's comments were considered. 

Author response

Thank you for your opinion about our work and for your comments.

Sincerely yours and on behalf of the authors,

Vera P. Tuguldurova
